# UNR-Explainer: Counterfactual Explanations for Unsupervised Node Representation Learning Models

**Hyunju Kang, Geonhee Han, Hogun Park**
Department of Artificial Intelligence
Sungkyunkwan University
Suwon, Republic of Korea
`{neutor,gunhee8178,hogunpark}@skku.edu`

## Abstract

Node representation learning, such as Graph Neural Networks (GNNs), has emerged as a pivotal method in machine learning. The demand for reliable explanation generation surges, yet unsupervised models remain underexplored in this regard. To bridge this gap, we introduce a method for generating counterfactual (CF) explanations in unsupervised node representation learning. We identify the most important subgraphs that cause a significant change in the $k$-nearest neighbors of a node of interest in the learned embedding space upon perturbation. The $k$-nearest neighbor-based CF explanation method provides simple, yet pivotal, information for understanding unsupervised downstream tasks, such as top-$k$ link prediction and clustering. Consequently, we introduce **UNR-Explainer** for generating expressive CF explanations for **U**nsupervised **N**ode **R**epresentation learning methods based on a Monte Carlo Tree Search (MCTS). The proposed method demonstrates superior performance on diverse datasets for unsupervised GraphSAGE and DGI. Our codes are available at `https://github.com/hjkng/unrexplainer`.

## 1 Introduction

Unsupervised node representation learning encodes networks into low-dimensional vector spaces without relying on labels. Prior studies (Grover & Leskovec, 2016; Kipf & Welling, 2017; Xu et al., 2019; Veličković et al., 2018; Xu et al., 2021) have demonstrated state-of-the-art performance across diverse tasks. Notably, the top $k$-nearest nodes significantly impact downstream tasks such as link prediction, clustering (Wang et al., 2023), outlier detection (Goodge et al., 2022), and recommendation (Boytsov et al., 2016). The growing prevalence of these models in real-world applications has led to an increased demand for understanding their outputs. Particularly, understanding why a node occupies a specific position is pivotal, shedding light on both learned embeddings and related tasks. Top-$k$ link prediction, for instance, predicts potential future links by considering the top-$k$ nearest unobserved nodes to the target node in the embedding space. Local clustering in Wang et al. (2023) efficiently approximates global clusters by categorizing top-k nearest nodes into homogeneous clusters.

Despite the importance of the top-$k$ nearest nodes in unsupervised representation learning, recent studies on explainability have largely overlooked them. While existing research (Yuan et al., 2020; Ying et al., 2019; Luo et al., 2020; Zhang et al., 2023) has mainly focused on explaining graph representation learning, relying on class labels, this method often falls short in unsupervised settings. Among the few studies that attempt to explain embedding vectors, Liu et al. (2018) utilize a hierarchical clustering model to represent learned embeddings in a taxonomy, providing a comprehensive view of the structure but limited in explaining individual nodes. TAGE (Xie et al., 2022) provides explanations for each instance by identifying subgraphs with high mutual information with the learned embeddings. However, TAGE is limited in its capacity for counterfactual reasoning as it doesn't optimize for counterfactual properties.

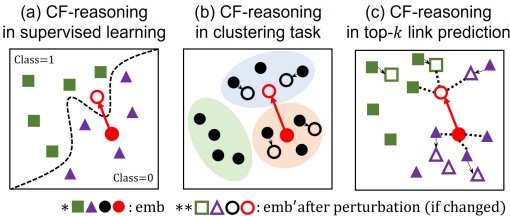

Figure 1: Scenarios for counterfactual (CF) reasoning in downstream tasks. Best viewed in color.

Counterfactual (CF) reasoning methods (Lucic et al., 2022; Lin et al., 2021; Bajaj et al., 2021; Tan et al., 2022; Ma et al., 2022) offer human-interpretable explanations by addressing the question "If certain edges had not occurred, would the prediction label $Y$ have remained unchanged?" Despite active research on CF reasoning in supervised settings for graph domains, its application in unsupervised settings remains underexplored. To explain a node of interest in the embedding space, we employ CF reasoning in unsupervised settings by minimally perturbing edges crucial for the current top-$k$ nearest nodes, yielding disparate results. Figure 1-(a) illustrates general CF reasoning in supervised settings, identifying edges causing label changes. However, this method is inapplicable in unsupervised settings due to the absence of class labels and decision boundaries. By utilizing top-$k$ nearest nodes, we can explain learned embedding vectors and related tasks in unsupervised settings. For instance, Figure 1-(b) depicts CF-reasoning in a clustering task, where perturbing pertinent edges change the node's local neighborhood in the embedding space, potentially changing cluster assignments. Similarly, Figure 1-(c) illustrates CF-reasoning in a top-$k$ (here, $k = 3$) link prediction task, where identifying edges that significantly affect top-$k$ neighbors may yield different results in embedding-based link prediction tasks.

To provide the CF reasoning in unsupervised learning, we introduce UNR-Explainer for unsupervised node representations (Hamilton et al., 2017; Veličković et al., 2018). Firstly, we define the CF explanation for unsupervised models by measuring the change of the top-$k$ nearest neighboring nodes after perturbation, formulating it as *Importance*. To identify the explanatory subgraph, we employ Monte Carlo Tree Search (MCTS) (Świechowski et al., 2022) tailored for our problem, inspired by Random Walk with Restart (Pan et al., 2004). Our novel subgraph traversal approach offers several advantages: 1) The subgraph by UNR-Explainer is sparse yet efficient as CF explanations. 2) UNR-Explainer finds more expressive subgraphs than existing MCTS methods by alleviating search bias, as elucidated in Theorem 1. UNR-Explainer provides meaningful CF explanations for unsupervised settings as demonstrated by extensive experiments. Given the significance of top-$k$ nearest nodes in downstream tasks, our method facilitates the exploration of significant factors affecting embedding vectors, thereby expanding the applicability of CF reasoning in unsupervised learning.

## 2 RELATED WORKS

**XAI for unsupervised models.** While active studies have been conducted on explaining supervised models, the interest in XAI for unsupervised models (Montavon et al., 2022; Crabbé & van der Schaar, 2022) has just ignited. In Montavon et al. (2022), neuralization-propagation (NEON) is proposed to explain k-means clustering and kernel density estimation using layer-wise relevance propagation. On the other hand, the authors in Crabbé & van der Schaar (2022) propose a method to explain the representation vectors without labels, providing a label-free importance score function to highlight the important features and examples. However, both approaches are limited to apply to graph-structure data without consideration of the relationship between instances.

**XAI for unsupervised node representation.** Recent studies (Liu et al., 2018; Xie et al., 2022; Park & Neville, 2023) have explored explainability for unsupervised node representation learning models on graphs. Liu et al. (2018) propose a method to explain the learned network embedding by describing the inherent hierarchical structures through a taxonomy using the hierarchical clustering model. This approach converts the representation vector into a taxonomy for a comprehensive view but is limited to explaining the prediction of a node of interest. On the other side, TAGE (Xie et al., 2022) explains individual nodes in the form of a subgraph sharing high mutual information with the embedding vector. Consequently, the explanation from the TAGE could expand to various downstream tasks predicting node or graph labels by simply updating the gradient of the class label in the downstream model. However, TAGE has a limitation in providing counterfactual explanations by minimal perturbation to change the class label. GRAPH-wGD (Park & Neville, 2023) is also relevant to our proposed method, while it aims to generate global explanations for learned models.

**Counterfactual explanations.** Most counterfactual reasonings for graphs explain predictions in supervised learning. Studies such as CF-GNNExplainer (Lucic et al., 2022), RCExplainer (Bajaj et al., 2021), and CF$^2$ (Tan et al., 2022) remove the important edges to emphasize the significant impact of edges in input on the prediction of class labels. CLEAR (Ma et al., 2022), and GCFExplainer (Huang et al., 2023) encourage the counterfactual result toward desired labels by adding new nodes or edges and removing the existing ones simultaneously. The mentioned methods are limited to applying unsupervised models, for instance, CF-GNNExplainer (Lucic et al., 2022) as a pioneer of counterfactual explanation for the graph domain employs its counterfactual loss which is limited to applying unsupervised models. RCExplainer (Bajaj et al., 2021) leverages the decision boundaries to tackle the robustness of counterfactual explanations, but the decision region heavily relies on class labels. In the same manner, other explainability methods make it difficult to exclude the information from the class labels. Contrasting with existing XAI methods in supervised learning, UNR-Explainer is designed to generate counterfactual explanations in unsupervised learning contexts without relying on labels.

## 3 Problem formulation

### 3.1 Node representation learning

**Notation** An input graph $\mathcal{G} = \{\mathcal{V}, \mathcal{E}, \mathcal{W}\}$ has a set of vertices $\mathcal{V} = \{v_0, ..., v_N\}$ with the node feature matrices $\mathbf{X} \in \mathbb{R}^{N \times d}$ and a set of edges $\mathcal{E} = \{(v_i, v_j)|v_i, v_j \in \mathcal{V}\}$ with a edge's weight set $\mathcal{W} = \{w_{(v_i, v_j)} \mid w_{(v_i, v_j)} \in [0, 1], (v_i, v_j) \in \mathcal{E}\}$. When the node $v$ has edges as $\mathcal{E}_v = \{(v_v, v_u)|(v_i, v_j) \in \mathcal{E}\}$, the neighboring nodes of the node $v$ is expressed as $u \in \mathcal{N}_v$.

**Node representation learning** such as GraphSAGE (Hamilton et al., 2017) aims to extract representation vectors of nodes by aggregating its attributes and information that is sampled from the local neighborhood. A trained unsupervised node representation model for an input graph $\mathcal{G}$ are represented as $f_{\text{unsup}}(\mathcal{G}) = \mathbf{emb}$. For a node $v \in \mathcal{V}$, the representation vector is given by $f_{\text{unsup}}(\mathcal{G}, v) = \mathbf{emb_v}$. It is calculated by stacking the embedding vector $\mathbf{h_v}$ for a node $v$ following as $\mathbf{h_v} = \sigma(\mathbf{M_{self}} \mathbf{x_v} + \mathbf{M_{agg}} \frac{1}{|\mathcal{N}_v|} \sum_{u \in \mathcal{N}_v} \mathbf{x_u})$ where $\mathbf{M_{self}}$ and $\mathbf{M_{agg}}$ are learnable parameter matrices. In unsupervised settings, it optimizes that nearby nodes are close and distant nodes are far apart in the embedding space by defining $k$-hop neighbors as positive samples and other nodes as negative ones.

### 3.2 Counterfactual Explanation for Unsupervised Node Representation learning

Given $f_{unsup}(\mathcal{G}, v) = \mathbf{emb_v}$, a counterfactual reasoning aims to identify an explanation $\mathcal{G}_s = \{\mathcal{V}_s, \mathcal{E}_s, \mathcal{W}_s\} \subset \mathcal{G}$ such that removing or weakening edges in $\mathcal{E}_s$ from $\mathcal{G}$ to form $\mathcal{G}' = \{\mathcal{V}, \mathcal{E} - \mathcal{E}_s, \mathcal{W} - \mathcal{W}_s\}$ results in $\mathbf{emb_v} \neq \mathbf{emb'_v}$ as $f_{unsup}(\mathcal{G}', v) = \mathbf{emb'_v}$ while minimizing the difference between $\mathcal{G}$ and $\mathcal{G}'$. The counterfactual explanation $\mathcal{G}_s$ for an unsupervised node representation learning is inherently difficult to define the counterfactual property as $\mathbf{emb_v} \neq \mathbf{emb'_v}$ being subject to $f_{unsup}(\mathcal{G}, v) = \mathbf{emb_v}$ and $f_{unsup}(\mathcal{G}', v) = \mathbf{emb'_v}$. It is problematic because any edge is possible to be the trivial counterfactual explanation after perturbation causing $\mathbf{emb_v} \neq \mathbf{emb'_v}$. Thus, defining the meaningful difference between two embedding vectors is necessary to provide a relevant counterfactual explanation for the target node $v$ of interest. Since top-$k$ nearest neighboring nodes are critical to downstream tasks, we exploit top-$k$ nearest neighboring nodes to define the counterfactual explanation for the unsupervised model. This approach is well aligned with models such as GraphSage and node2vec (Grover & Leskovec, 2016; Hamilton et al., 2017) since their objective function optimizes the embedding vector in which similar nodes are located relatively close while dissimilar ones are far away. Thus, the top-$k$ nearest neighboring nodes share similar features for related downstream tasks. Therefore, we utilize the top-$k$ nearest neighboring nodes to define the counterfactual property for unsupervised representation learning models as follows:

**Definition 1** *Given $f_{unsup}(\mathcal{G}, v) = \mathbf{emb_v}$ and $f_{unsup}(\mathcal{G}', v) = \mathbf{emb'_v}$, a hyper-parameter $k$, and a function $kNN(\mathbf{emb}, v, k)$, a counterfactual property for the node $v$ as $\mathbf{emb_v} \neq \mathbf{emb'_v}$ is satisfied when $kNN(\mathbf{emb}, v, k) \neq kNN(\mathbf{emb'}, v, k)$.*

### 3.3 MEASURING *Importance* FOR COUNTERFACTUAL EXPLANATIONS

Upon the Definition 1, we define a measure to quantify the *Importance* of the subgraph $\mathcal{G}_s$ as counterfactual explanation below:

$$Importance(f_{\text{unsup}}(\cdot), \mathcal{G}, \mathcal{G}_s, v, k) = |set(kNN(f_{\text{unsup}}(\mathcal{G}), v, k)) - set(kNN(f_{\text{unsup}}(\mathcal{G} - \mathcal{G}_s), v, k))|/k, \quad (1)$$

where $kNN$ is found based on the Euclidean distance. Using *Importance*, we define a counterfactual explanation $\mathcal{G}_s$ for the embedding vector in unsupervised node representation learning as below:

**Definition 2** *Given $f_{unsup}(\mathcal{G}, v) = \mathbf{emb_v}$, a counterfactual reasoning aims to identify a subgraph $\mathcal{G}_s = \{\mathcal{V}_s, \mathcal{E}_s, \mathcal{W}_s\} \subset \mathcal{G}$ such that removing or weakening edges in $\mathcal{E}_s$ from $\mathcal{G}$ to form $\mathcal{G}' = \{\mathcal{V}, \mathcal{E} - \mathcal{E}_s, \mathcal{W} - \mathcal{W}_s\}$ maximizes the Importance while minimizing the difference between $\mathcal{G}$ and $\mathcal{G}'$.*

If the subgraph $\mathcal{G}_s$ is critical to the target node's node embedding, the effect of the perturbation must be significant based on the counterfactual assumption. Hence, the subgraph $\mathcal{G}_s$ is employed as the explanation for the target node's node embedding. If the *Importance* is 0, we observe no effect on the top-$k$ neighboring nodes in the updated embedding space. Meanwhile, when the *Importance* is 1, all surrounding neighbors are changed, fully satisfying the counterfactual property. We note that we leverage an LSH hashing (Shrivastava & Li, 2014) to obtain the nearest neighbors efficiently. To obtain the perturbed $\mathcal{G}'$, we employ the perturbation method which weakens the weight of the input graph edges equivalent to the edges of the subgraph. An overview of *Importance* function is provided in Algorithm 2 in Appendix B. Additionally, we theoretically analyze the upper bound of *Importance* upon GraphSAGE in Theorem 2 in Appendix C.2.

## 4 OUR PROPOSED METHOD

Our goal is to find an explanation subgraph $\mathcal{G}_s$ as a counterfactual explanation with the highest *Importance* score to change the top-$k$ neighbors in embedding space while maintaining the minimum edge size $|\mathcal{G}_s|$. It is defined as

$$\underset{|\mathcal{G}_s|}{\arg\min} \underset{\mathcal{G}_s}{\max} Importance(f_{unsup}(\cdot), \mathcal{G}, \mathcal{G}_s, v, k). \quad (2)$$

Due to the exponential number of existing subgraphs, an efficient traversal method is required. Thus, we leverage the Monte Carlo Tree Search (MCTS) (Sutton & Barto, 2018) which is one of reinforcement learning and well-known to outperform in large search spaces (Silver et al., 2017). MCTS utilizes a search tree in which a series of actions is selected from a root to a leaf node resulting in the maximum reward according to the pre-defined reward function. We tailor the MCTS to suit our problem setting by applying the *Importance* as the reward to obtain the explanation subgraph $\mathcal{G}_s$. This method brings the advantages of not only searching subgraphs efficiently thanks to the reinforcement algorithm but also searching the subgraph sparse yet expressive.

### 4.1 SUBGRAPH TRAVERSAL METHOD

We initialize a search tree in which each node $N_i$ has properties as $\{S(N_i), A(N_i)\}$ where $S(N_i)$ denotes a state and $A(N_i)$ represents a set of actions. The node $N_i$ in the tree corresponds to the node $v \in \mathcal{V}$ in the input graph $\mathcal{G}$, and $S(N_i)$ indicates an index of the node $v$. The edges in the tree mean actions $a \in A(N_i)$. The action is to select one node in the search tree equivalently meaning that we add a new edge or node gradually in the candidate subgraph. To start in the initial search tree, we set a root node $N_0, S(N_0) = v$ as the index of the node $v$ to explore subgraphs around the target node $v$. At each iteration $t$, the algorithm travels from the root node $N_0$ to a leaf node $N_i$, resulting in the trajectory $\tau_t = \{(N_0, a_1), ..., (N_i, a_j)\}$. Then, we can transform the path $\tau_t$ to a subgraph $\mathcal{G}_s = \{\mathcal{V}_s, \mathcal{E}_s, \mathcal{W}_s\}, \mathcal{V}_s = \{S(N_0), ..., S(N_i)\}, \mathcal{E}_s = \{(S(N_i), S(N_k))|S(N_i), S(N_k) \in \mathcal{V}_s, a_j = N_i, a_{j+1} = N_k, a_0 = N_0\}$ defined as a function $Convert(\tau) = \mathcal{G}_\tau$. Consequently, searching the subgraph $\mathcal{G}_s$ by searching the path via the MCTS algorithm is possible. A pair $(N_i, a_j)$ of node $N_i$ and action $a_j$ stores statistics $\{C(N_i, a), Q(N_i, a), L(N_i, a)\}$ which are described as following: **1)** $C(N_i, a)$ is the number of visits for selecting the action $a \in A(N_i)$ at the node

---

**Algorithm 1** UNR-Explainer with restart

---

1: **Input**: trained model $f_{unsup}(\cdot)$, input graph $\mathcal{G}$, $kNN$ parameter $k$, target node $v$, exploration term $\lambda$, a probability $p \sim Uniform(0, 1)$ and restart probability $p_{restart}$
2: **Initialization**: Initialize $\mathcal{G}_l = \{\}$, step $t = 0$, and set a root node $N_0, S(N_0) = v$
3: **while** termination condition **do**
4:     $curNode \leftarrow N_0$; a trajectory $\tau = \{\}$; $t += 1$; Expand the child nodes of $curNode$
5:     Select $a \in A(curNode)$ by $UCB(curNode, a)$ in Eq.(3)
6:     **while** $curNode$ is not leaf node **do**
7:         **if** $p \leq p_{restart}$ **then**
8:             $curNode \leftarrow N_0$; Append a pair $(N_0, a)$ into the trajectory $\tau$
9:             Obtain $a^*$ using Eq.(3) and select randomly action $a \in A(N_0) \smallsetminus a^*$
10:            $curNode \leftarrow N_i$ by action $a$
11:         **end if**
12:         Select $a \in A(curNode)$ by $UCB(curNode, a)$ in Eq.(3)
13:         Append a pair $(curNode, a)$ into the trajectory $\tau$
14:         $curNode \leftarrow N_i$ by action $a$
15:     **end while**
16:     **if** the leaf node has no child **then**
17:         Expand the child nodes of $curNode$
18:         Select $a \in A(curNode)$ by $UCB(curNode, a)$ in Eq.(3)
19:         Append a pair $(curNode, a)$ into the trajectory $\tau$
20:     **end if**
21:     Calculate the reward by $R(\tau)$
22:     Backpropagate the reward and visit counts to $(N_i, a) \in \tau$
23:     $\mathcal{G}_\tau \leftarrow Convert(\tau)$; Append $\mathcal{G}_\tau$ into the list $\mathcal{G}_l$
24: **end while**
25: **Return** $\mathcal{G}_s$ from $\mathcal{G}_l$ using Eq.(2)

---

$N_i$. **2)** $Q(N_i, a)$ is an action value that is calculated as the maximum value in a set of rewards $L(N_i, a)$ aiming to obtain the highest important subgraph. **3)** $L(N_i, a)$ is a list of obtained rewards from reward function $R(\tau)$ in every iteration when $(N_i, a) \in \tau$. **4)** $R(\tau)$ is the reward function to calculate *Importance* as $R(\tau) = Importance(f_{\text{unsup}}(\cdot), \mathcal{G}, \mathcal{G}_\tau, k, v)$. We obtain the subgraph $\mathcal{G}_s$ from $Convert(\tau) = \mathcal{G}_\tau$. Utilizing the statistics, the algorithm searches the path with the highest *Importance* based on the upper confidence boundary (UCB) in (Świechowski et al., 2022) as $UCB(N_i, a) = Q(N_i, a) + \lambda\, P\, \sqrt{\frac{ln(C(N_0, a))}{C(N_i, a)}}$ :

$$a^* = \arg\max_a UCB(N_i, a), a \in A(N_i) \tag{3}$$

While the term $Q(N_i, a)$ promotes choosing the action with a higher reward, the latter term relating to $C(N_i, a)$ encourages exploration by choosing the less visited action. A hyperparameter $\lambda$ controls the magnitude of exploration to balance exploitation and exploration. Additionally, the term $P$ intends to provide useful information for the guidance of exploration. In the case of the term $P = 1$ as constant, UCB does not utilize the additional guidance. Further, we discuss the benefit of our setting compared to other employments of MCTS and prove the efficiency of our exploration method through the ablation study in Table 4.

### 4.2 THE PROPOSED UNR-EXPLAINER

The vanilla MCTS algorithm in our setting has an issue of low expressiveness in the selection step in the aspect of the UCB formula theoretically. Because the existing UCB-based search policy tends to generate the subgraph in a depth-first manner rather than in a breadth-first manner, the previously appeared node is seldom selected by the UCB-based formula, leading not to explore close hop neighboring nodes even its importance. Based on an empirical assumption demonstrated experimentally in studies (Bajaj et al., 2021; Yuan et al., 2021), we present a theoretical analysis of the expressiveness limitation inherent in the vanilla MCTS algorithm as follows.

**Theorem 1** *Let be an action $a_j$ at an arbitrary node $N_i$ in a trajectory $\tau_t = \{(N_0, a_1), ..., (N_i, a_j)\}$, resulting in a next node $N_k$. Suppose $a_1, a_2 \in A(N_k)$ where the action $a_1$ leads to a node $N_m$ with a state function $S(N_m) \neq S(N_i)$, and the action $a_2$ leads to a node $N_n$ with $S(N_n) = S(N_i)$. Then, we have*

$$UCB(N_k, a_1) \geq UCB(N_k, a_2) \qquad (4)$$

We provide a proof and detailed analysis of Theorem 1 in Appendix C.1. To ensure that the resulting subgraphs of the traversal are more expressive and accurate, we introduce a new selection policy in the following section. The overview of the algorithm is described in Algorithm 1.

**Selection with a new policy.** At every iteration, a restart probability $p_{restart} \in [0, 1], p \sim Uniform(0, 1)$ is sampled. When $p < p_{restart}$, it returns to the root node and randomly explores another path $a \in A(N_0) \setminus a^*$ excluding the optimal one from the root. Thus, it promotes more connections centered around the target node. Implementing the restart probability as a new selection policy has several advantages: **1)** It reduces the bias towards deep-first traversal, increasing the likelihood of generating diverse candidate subgraphs. **2)** It helps to prevent getting stuck on an isolated island or infinite iteration. When $p \geq p_{restart}$, the algorithms select the child nodes with the higher UCB score in Eq. (3). The selection continues until it reaches the leaf node. It is described in line from 5 to 15 of Algorithm 1.

**Expansion.** When arriving at the leaf node, the tree expands if the number of visits to the leaf node is more than one. The tree adds the child nodes from the neighboring nodes of the current node's state in the input graph $\mathcal{G}$. However, considering all neighbor nodes as child nodes causes exponential search space. Therefore, we randomly select nodes from the neighbors of the leaf nodes in the input graph $\mathcal{G}$ to compute them more efficiently, setting the expansion number $E$ to be equal to the average degree of the input graph $\mathcal{G}$. It is described in line from 16 to 20 of Algorithm 1.

**Simulation.** We transform the obtained trajectory $\tau$ to the subgraph $\mathcal{G}_\tau$. As The generated subgraph $\mathcal{G}_\tau$ is a candidate of the important subgraph stored in a list $\mathcal{G}_l$, we measure the *Importance* of the subgraph $\mathcal{G}_\tau$ for the reward. It is described in line 21 of Algorithm 1.

**Backpropagation.** Finally, we update the reward and the number of visits to the statistics of related nodes in the trajectory. The number of visits $C(N_i, a)$ is renewed for $C(N_i, a) + 1$. Then, we append the obtained reward to a list of rewards $L(N_i, a)$. Consequently, the action value $Q(N_i, a)$ is updated by $max(L(N_i, a))$. These updated statistics are exploited for the next selection to explore the subgraph with higher *Importance*. It is described in line 22 of Algorithm 1.

**Termination condition.** Our subgraphs have two main properties as the counterfactual explanation. First, the importance is close to 1, affecting all the top-$k$ neighbors after perturbation. Second, the size of the subgraph is minimal. To achieve the goal, we explore the subgraph setting the terminal condition as *Importance* == 1.0 with limited but enough steps $T$. When the algorithm ends after meeting the condition, we select the subgraph with a minimum size of the subgraph among the one with maximum *Importance*. As a result, we effectively search the explanatory subgraphs satisfying the properties of counterfactual explanations.

## 5 EXPERIMENTS

### 5.1 DATASETS

The experiments are conducted on three synthetic datasets (Ying et al., 2019; Luo et al., 2020) and three real-world datasets from PyTorch-Geometrics (Fey & Lenssen, 2019). The synthetic datasets (BA-Shapes, Tree-Cycles, and Tree-Grid (Ying et al., 2019)) are used, containing network motifs such as houses, cycles, and grid-structure motifs respectively. These motifs serve as the ground truth of the datasets to evaluate the explanation method (Yuan et al., 2020). The real-world datasets (Cora, CiteSeer, and PubMed (Grover & Leskovec, 2016)) are citation networks commonly used in graph domain tasks. Each node indicates a paper and each edge notes a citation between papers. The features of the nodes are represented as bag-of-words of the papers and each node is labeled as the paper's topic. Additionally, we exploit the NIPS dataset from Kaggle to employ the case study of our method. The statistics of the datasets are reported in Table 7 in the Appendix.

Table 1: Evaluation of CF explanations on synthetic datasets in unsupervised settings w.r.t. ground-truth. Prc, Rcl, and Impt indicate Precision, Recall, and Importance. The best performances on each dataset are shown in **bold**.

| | BA-Shapes | | | | Tree-Cycles | | | | Tree-Grids | | | |
|---|---|---|---|---|---|---|---|---|---|---|---|---|
| Methods | Prc ↑ | Rcl ↑ | Impt ↑ | Size ↓ | Prc ↑ | Rcl ↑ | Impt ↑ | Size ↓ | Prc ↑ | Rcl ↑ | Impt ↑ | Size ↓ |
| 1hop-2N | 0.934 | 0.156 | 0.492 | **1.0** | 0.900 | 0.150 | 0.277 | **1.0** | **0.965** | 0.161 | 0.683 | 1.0 |
| 1hop-3N | **0.943** | **0.314** | 0.817 | 2.0 | 0.895 | 0.298 | 0.994 | 2.0 | 0.959 | 0.320 | 0.633 | 2.0 |
| $k$-NN graph | 0.025 | 0.012 | 0.392 | 1.8 | 0.017 | 0.006 | 0.148 | 1.5 | 0.013 | 0.004 | 0.337 | 1.1 |
| RW-G | 0.890 | 0.263 | 0.597 | 1.8 | 0.862 | 0.302 | 0.591 | 2.1 | 0.925 | 0.459 | 0.834 | 3.4 |
| RW-G w/ Restart | 0.881 | 0.268 | 0.683 | 1.9 | 0.840 | 0.300 | 0.641 | 2.1 | 0.917 | 0.496 | 0.922 | 3.6 |
| Taxonomy induction | 0.731 | 0.310 | 0.292 | 1.8 | 0.765 | 0.298 | 0.454 | 1.8 | 0.377 | 0.140 | 0.369 | 0.9 |
| TAGE | 0.422 | 0.225 | 0.220 | 2.4 | 0.290 | 0.230 | 0.166 | 1.7 | 0.415 | 0.360 | 0.260 | 2.3 |
| **UNR-Explainer** | 0.923 | 0.288 | **1.000** | 1.9 | **0.903** | **0.324** | **1.000** | 2.1 | 0.943 | **0.519** | **0.994** | 3.6 |

## 5.2 BASELINE METHODS

In unsupervised settings, we compare **UNR-Explainer** with seven baselines as follows: 1) **1hop-2N** is a subgraph of an ego network in which edges are randomly removed until the number of nodes is two 2) **1hop-3N** is a subgraph of an ego network in which edges are randomly removed until the number of nodes is three. 3) $k$**-NN graph** is a subgraph whose nodes and edges are top-5 nearest neighbors in the embedding space. 4) **RW-G** generates a subgraph by the random walk algorithm by traveling neighboring nodes from the target node $v$. For a fair comparison, the number of nodes and edges in the graph is mostly equal to one from our proposed method. 5) **RW-G w/ Restart** are generated in the same way as RW-G but applying random walk with restart. 6) **Taxonomy induction** (Liu et al., 2018) outputs the clusters as the global explanation of the embedding vector. We expect that the subgraphs in the cluster can be utilized as a local explanation. 7) **TAGE** (Xie et al., 2022) generates a subgraph with high mutual information of the learned embedding vector.

## 5.3 EVALUATION METRICS

We evaluate explanation subgraphs in the counterfactual aspect by the following metrics: 1) **Importance** of the explanation subgraph $\mathcal{G}_s$ is utilized to measure the impact in the embedding space. 2) **Size** (Tan et al., 2022) shows how the explanation is effective with minimal perturbation on synthetic and real-world datasets. 3) **Precision / Recall** (Tan et al., 2022) on average w.r.t ground truth are measured on the synthetic datasets. 4) **Validity** (Verma et al., 2020) is commonly used to effectiveness related to $f(\mathcal{G}) \neq f(\mathcal{G}')$ in supervised learning. By utilizing *Importance*, we calculated the averaged validity for the unsupervised model as $Validity(\mathcal{G}_s) = \mathbb{1}\big[Importance(f_{unsup}(\cdot), \mathcal{G}, \mathcal{G}_s, v, k) = 1.0\big]$. 5) **Probability of Necessity as PN** (Tan et al., 2022) in top-$k$ link prediction tasks are used to measure how predictions as $Hit@5$ are changed to demonstrate the effectiveness of the explanations. 6) **Homogeneity** represents the percentage that the top-$k$ neighbors of a node of interest belong to the same cluster of the node of interest in the node embedding space after perturbing the generated explanation for each baseline. Original Homogeneity in the learned embedding space before perturbation is expressed as the input graph in Table 3. 7) $\Delta$ **Homogeneity** means the difference between the $Homogeneity$ of Input graph and the $Homogeneity$ of each baseline.

## 5.4 RQ1: PERFORMANCE OF UNR-EXPLAINER AND OTHER BASELINE MODELS

We demonstrate the performance of UNR-Explainer showing our explanation graphs change the top-$k$ nearest nodes and impacts related downstream tasks in node classification, link prediction, and clustering in unsupervised settings in Table 1, 2, and 3. From Table 1 on synthetic datasets, we demonstrate that UNR-Explainer describes the ground truth, showing the highest $Recall$ and $Importance$ among baselines on BA-Shapes and Tree-Cycles datasets. In the experiment, 1hop-3N demonstrates the highest $Precision$ on both the BA-Shapes and Tree-Cycles datasets, and 1hop-2N exhibits the highest $Precision$ on the Tree-Grids dataset. Regarding the BA-Shape dataset, there are 80 house-structured motifs, each comprising 5 nodes as the ground truth, and only a few edges in these motifs are connected to the base BA graph. When we generate explanations as 1hop-2N and 1hop-3N for nodes within these house motifs, the precision tends to be high. This is due to the increased probability that the explanation will include another node from the house-structured motif, thereby aligning with the ground truth. Similarly, for other synthetic datasets constructed in the same manner, 1hop-2N and 1hop-3N exhibit high precision but low recall. $k$-NN graph in

Table 2: Evaluation of CF explanations on real-world datasets in unsupervised settings. Vld, Impt, and PN indicate Validity, Importance, and Probability of Necessity. The best performances on each dataset are shown in **bold**.

| Methods | Cora | | | | CiteSeer | | | | PubMed | | | |
|---|---|---|---|---|---|---|---|---|---|---|---|---|
| | Vld ↑ | Impt ↑ | PN ↑ | Size ↓ | Vld ↑ | Impt ↑ | PN ↑ | Size ↓ | Vld ↑ | Impt ↑ | PN ↑ | Size ↓ |
| 1hop-2N | 0.210 | 0.465 | 0.006 | **1.0** | 0.340 | 0.551 | 0.020 | **1.0** | 0.485 | 0.623 | 0.000 | **1.0** |
| 1hop-3N | 0.428 | 0.659 | 0.021 | 2.0 | 0.503 | 0.714 | 0.035 | 1.6 | 0.639 | 0.724 | 0.000 | 1.5 |
| $k$-NN graph | 0.253 | 0.593 | 0.019 | 3.3 | 0.243 | 0.609 | 0.028 | 3.3 | 0.022 | 0.090 | 0.000 | **1.0** |
| RW-G | 0.265 | 0.544 | 0.019 | 3.6 | 0.393 | 0.643 | 0.024 | 2.7 | 0.478 | 0.631 | 0.027 | 2.0 |
| RW-G w/ Restart | 0.327 | 0.614 | 0.019 | 3.8 | 0.453 | 0.696 | 0.029 | 2.8 | 0.514 | 0.662 | 0.029 | 3.1 |
| Taxonomy induction | 0.243 | 0.548 | 0.023 | 9.7 | 0.236 | 0.528 | 0.037 | 4.9 | 0.035 | 0.061 | 0.000 | 1.8 |
| TAGE | 0.232 | 0.482 | 0.016 | 5.4 | 0.254 | 0.487 | 0.031 | 3.7 | 0.082 | 0.268 | 0.012 | 11.8 |
| **UNR-Explainer** | **0.911** | **0.960** | **0.051** | 3.8 | **0.778** | **0.910** | **0.054** | 2.8 | **0.847** | **0.903** | **0.034** | 3.7 |

Table 3: Evaluation of CF explanations upon clustering tasks on real-world datasets. Hmg and $\Delta$ Hmg indicate Homogeneity and the change of Homogeneity.

| Methods | Cora | | CiteSeer | | PubMed | |
|---|---|---|---|---|---|---|
| | Hmg ↓ | $\Delta$ Hmg ↑ | Hmg ↓ | $\Delta$ Hmg ↑ | Hmg ↓ | $\Delta$ Hmg ↑ |
| Input graph | 0.866 | - | 0.863 | - | 0.697 | - |
| 1hop-2N | 0.695 | 0.171 | 0.564 | 0.298 | 0.372 | 0.325 |
| 1hop-3N | 0.520 | 0.347 | 0.401 | 0.462 | 0.299 | 0.398 |
| $k$-NN graph | 0.621 | 0.245 | 0.629 | 0.234 | 0.636 | 0.061 |
| RW-G | 0.646 | 0.220 | 0.487 | 0.376 | 0.364 | 0.333 |
| RW-G w/ Restart | 0.594 | 0.272 | 0.441 | 0.422 | 0.331 | 0.366 |
| Taxonomy induction | 0.598 | 0.268 | 0.595 | 0.268 | 0.579 | 0.118 |
| TAGE | 0.627 | 0.240 | 0.569 | 0.294 | 0.555 | 0.142 |
| **UNR-Explainer** | **0.315** | **0.551** | **0.272** | **0.591** | **0.243** | **0.454** |

the embedding space is limited to constructing the subgraph valid in the input graph since graph neural networks encounter challenges preserving graph topology. RW-G w/ Restart demonstrates our proposed selection policy, showing higher recall than RW-G with a slightly larger size. Explanations from Taxonomy induction based on global clusters are not as sufficient as local CF explanations. On the Tree-Grid dataset, UNR-Explainer does not reach the ground truth due to the minimal constraint of the size. In the case of TAGE, it reaches the highest recall on Tree-Grids datasets but the largest size and the lowest $Importance$ without satisfying counterfactual measurements.

Table 2 shows the results on real-world datasets, where UNR-Explainer records the best score in all metrics except size. Explanations by UNR-Explainer impact most of the top-$k$ neighbors after perturbation, showing the highest $Validity$ and $Importance$ with a smaller size of subgraphs than Taxonomy induction and TAGE. When we apply the embedding vector into top-5 link prediction tasks, originally the accuracy of the prediction is $0.570, 0.575$, and $0.55$ respectively on Cora, CiteSeer, and Pubmed datasets. After the perturbation of each explanation, the predictions w.r.t $PN$ change the most in the case of UNR-Explainer due to the significant difference of top-$k$ neighbors. None of Taxonomy induction and TAGE optimize the counterfactual term, resulting in lower $PN$. On PubMed datasets, we sample 10% of nodes and evaluate the change of probability in node classification.

In Table 3, we demonstrate that explanatory graphs affect clustering tasks in unsupervised settings. When $k$ is equal to 20, the majority of the top-$k$ neighbors in the embedding space typically belong to the same cluster of a node of interest, as observed by the $homogeneity$ values of $0.866, 0.863$, and $0.697$ on the Cora, CiteSeer, and PubMed datasets, respectively. After perturbating of found explanations by UNR-Explainer, the $homogeneity$ of original top-$k$ neighbors drops significantly at the most among baselines, which means that a node of interest is no longer a part of the cluster but a different one as a counterfactual result. Hence, explanations by UNR-Explainer affect top-$k$ neighbors in the embedding but also the output of related downstream tasks.

## 5.5 RQ2: A CASE STUDY IN COMMUNITY DETECTION ON NIPS DATASETS

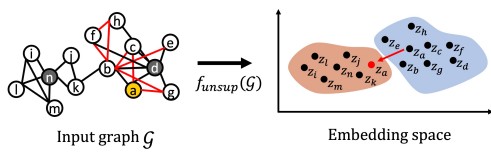

Input graph $\mathcal{G}$       Embedding space

Figure 2: Case study on NIPS datasets.

Our explanatory graphs reveal the important connections that impact the top-$k$ neighboring nodes in embedding space, particularly for here case study on a social network. We apply our method on a co-author network dataset of the NIPS conferences (Hamner, 2017). Upon applying Graph-SAGE in unsupervised settings, we extract signifi-

cant subgraphs using UNR-Explainer, illustrating its effectiveness in impacting a k-means clustering task. In Figure 2, a target node, highlighted as yellow node $a$ representing Caglar Gulcehr, is initially a part of the cluster of Yoshua Bengio, known for publications primarily related to deep learning and general machine learning, represented by node $d$. In this figure, node positions in the embedding space are determined through t-SNE learning (van der Maaten & Hinton, 2008). After perturbing the found explanation graph, illustrated by red edges, the node aligns with the cluster of Nicolas Heess, notable for publications primarily related to reinforcement learning, as represented by node $n$.

## 5.6 RQ3: ABLATION STUDY WITH OTHER MCTS-BASED SUBGRAPH TRAVERSAL METHODS

Table 4: Ablation study of the MCTS-based method on Cora dataset.

| Name | Time(s) | Impt | Size |
|---|---|---|---|
| MCTS | 11.30 | 0.942 | **3.43** |
| SubgraphX-1 | 176.30 | 0.936 | 28.49 |
| SubgraphX-2 | 146.02 | 0.937 | 25.66 |
| MCTS-Avg | 10.11 | 0.949 | 3.62 |
| MCTS-Prx1 | 9.85 | 0.951 | 3.71 |
| MCTS-Prx2 | 7.14 | 0.956 | 3.59 |
| MCTS-Prx3 | 10.55 | 0.953 | 3.60 |
| UNR-Explainer w/o r | 8.11 | 0.955 | 3.67 |
| **UNR-Explainer** | **4.63** | **0.960** | 4.5 |

We evaluate variants of the MCTS algorithm considering the design of the action, the expansion, the UCB-based formula, and the restart as shown in the result in Table 4. As evaluation metrics, *Importance*, explanation size, and the mean inference time (s) per node are measured. As a result, UNR-Explainer shows the best results considering the importance and inference time (s). In SubgraphX (Yuan et al., 2021), its action is to prune equally meaning to remove nodes. Since counterfactual explanations must be minimal with high validity, SubgraphX fails to find the subgraph particularly when it deals with the high-degree target node showing bigger size than other baselines. We note that Prx means proximity as the term $P$ for better guidance in Eq 3. However, none of the Prx helps guide the important subgraph. On the contrary, our proposed UNR-Explainer makes it possible to search the sparse yet expressive subgraph efficiently with high importance. Detailed information including the node's proximity (Prx) is written in Appendix D.3.

## 5.7 RQ4: PARAMETER SENSITIVITY

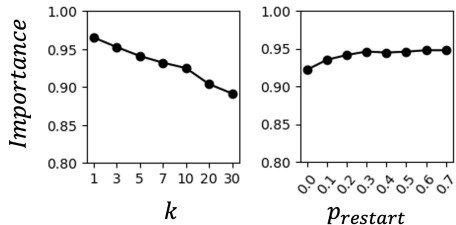

Figure 3: The study of parameter sensitivity for the number of the nearest neighbors as $k$ and the restart probability $p_{restart}$.

We evaluate the robustness of our model relating to crucial hyperparameters, such as the top-$k$ number of neighbors $k$ and the restart probability $p_{restart}$, using the GraphSAGE model on the Cora dataset. The results are shown in Figure 3. As the hyperparameter $k$ increases, the importance decreases. This is because the hyperparameter $k$ determines the range of the local region in the embedding space, and a larger region containing nodes further away from the target node is less affected. Furthermore, the importance increases until the restart probability $p_{restart}$ reaches 0.2, the value used in our model's setting. Additionally, we observe that the exploration term as $\lambda$ does not affect the models' performance much.

## 6 CONCLUSION

In this paper, we propose a counterfactual explanation method, UNR-Explainer, to explain the unsupervised node representation vector of a single node. We define $k$-nearest neighbor-based counterfactual explanation and propose a new selection policy to identify the important subgraph that prioritizes returning back to the target node to find multiple paths in the search tree. Our experimental results show that the important subgraphs generated by UNR-Explainer have higher local impacts in both unsupervised and supervised downstream tasks compared to other baseline methods.

# 7 ACKNOWLEDGMENT

We thank the anonymous reviewers for their valuable feedback and insights. Hogun Park is the corresponding author. This work was supported by the National Research Foundation of Korea (NRF) (2021R1C1C1005407); and by the Institute of Information & Communications Technology Planning & Evaluation (IITP) grant funded by the Korea government (MSIT): (No. 2019-0-00421, Artificial Intelligence Graduate School Program (Sungkyunkwan University)) and (No. RS-2023-00225441, Knowledge Information Structure Technology for the Multiple Variations of Digital Assets). This research was also supported by the Culture, Sports, and Tourism R&D Program through the Korea Creative Content Agency grant funded by the Ministry of Culture, Sports and Tourism in 2024 (Project Name: Research on neural watermark technology for copyright protection of generative AI 3D content, 25%).

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

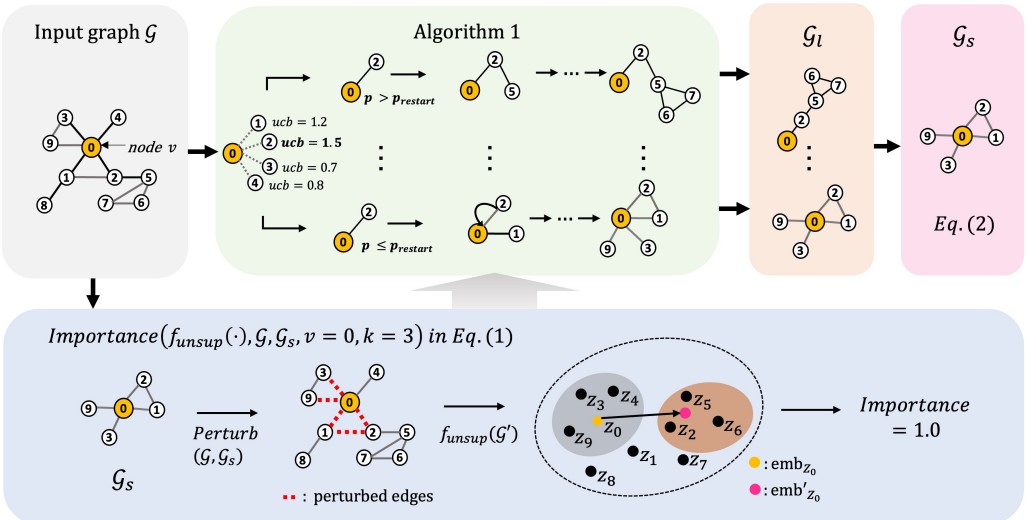

Figure 4: Overall procedure of UNR-Explainer to explain the unsupervised node representation embedding of node 0 (target node). Our proposed method is to search the most important subgraph $\mathcal{G}_s$ as the explanation for the target node. (Best viewed in color.)

## A    OVERVIEW OF UNR-EXPLAINER

The overview of UNR-Explainer is provided in Figure 4. Initially, the input $\mathcal{G}$ and a learned model $f_{unsup}(\cdot)$ are provided. These are used in the importance function described in Eq. (1) under Algorithm 1 to traverse the most contributive subgraph.

## B    THE ALGORITHM OF *Importance*

We describe the process of calculating the *Importance* for the subgraph $\mathcal{G}_s$ as detailed in Algorithm 2. First, the perturbation is to remove the edges of the subgraph $\mathcal{G}_s$ from the input graph $\mathcal{G}$. Subsequently, we construct the perturbed graph $\mathcal{G}_{perturbed}$ and obtain the $k$-nearest neighbors ($kNN$) for the target node $v$. Finally, the *Importance* of $\mathcal{G}_s$ is evaluated as the difference between the top-$k$ nearest neighbors in $\mathcal{G}$ and $\mathcal{G}_{perturbed}$.

---

**Algorithm 2** $Importance(f_{unsup}(\cdot), \mathcal{G}, \mathcal{G}_s, v, k)$

---

1: Input: embedding model $f_{unsup}(\cdot)$, input graph $\mathcal{G} = (\mathcal{V}, \mathcal{E}, \mathcal{W})$, subgraph $\mathcal{G}_s = (\mathcal{V}_s, \mathcal{E}_s, \mathcal{W}_s)$, target node $v$, the number of top-$k$ neighbors in $kNN$ as $k$
2: $\mathcal{G}_{perturbed} \leftarrow (\mathcal{V}, \mathcal{E} - \mathcal{E}_s, \mathcal{W})$
3: $\mathbf{emb} \leftarrow f_{unsup}(\mathcal{G})$
4: $\mathbf{emb'} \leftarrow f_{unsup}(\mathcal{G}_{perturbed})$
5: $O \leftarrow kNN(\mathbf{emb}, v, k)$  # get the top-$k$ neighbors of $v$ in $\mathbf{emb}$
6: $N \leftarrow kNN(\mathbf{emb'}, v, k)$  # get the top-$k$ neighbors of $v$ in $\mathbf{emb'}$
7: Return $|(set(O) - set(N)|/k$  # *Importance*

---

## C    THEORETICAL ANALYSIS

### C.1    EXPRESSIVENESS OF MCTS-BASED SUBGRAPH TRAVERSAL

We present a theoretical analysis of the expressiveness limitation inherent in the vanilla MCTS algorithm, specifically its restriction to searching for the optimal explanatory subgraph. We analyze the low expressiveness in the selection step in the aspect of the UCB formula theoretically. In the vanilla MCTS, generating the subgraph in a breadth-first manner requires a previously appeared

node as a newly expanded child node in the tree to return back to the root node. Yet, the previously appeared node is seldom selected by the UCB-based formula, as the action doesn't yield any gain due to no difference in $E_s$. Before presenting this analysis, we introduce an empirical assumption that has been demonstrated experimentally in studies (Bajaj et al., 2021; Yuan et al., 2021), as follows:

**Assumption 1** *The graph that is perturbed by more edges of $v \in \mathcal{V}$ results in changes to more components of $kNN(\mathbf{emb}, v, k)$ in the embedding space.*

Based on Assumption 1, we theoretically analyze the expressiveness of the vanilla MCTS subgraph traversal procedure in Theorem 1 and present proof as follows.

**Theorem 1** *Let be an action $a_j$ at an arbitrary node $N_i$ in a trajectory $\tau_t = \{(N_0, a_1), ..., (N_i, a_j)\}$, resulting in a next node $N_k$. Suppose $a_1, a_2 \in A(N_k)$ where the action $a_1$ leads to a node $N_m$ with a state function $S(N_m) \neq S(N_i)$, and the action $a_2$ leads to a node $N_n$ with $S(N_n) = S(N_i)$. Then, we have*

$$UCB(N_k, a_1) \geq UCB(N_k, a_2) \tag{5}$$

*Proof.* When the algorithm selects an action $(N_k, a_1)$ or $(N_k, a_2)$ using Equation (3) at the node $N_k$, we assume that the number of visit counts is identical as $C(N_k, a_1) = C(N_k, a_2)$. Then only term $Q(N_k, a_\forall)$ matters in Equation (3). To calculate the *Imporance* as the reward, the trajectory $\tau_t$ is converted into the subgraph $\mathcal{G}_{\tau_t} = \{\mathcal{V}_{\tau_t}, \mathcal{E}_{\tau_t}\}$ using the function $Convert(\tau_t)$. Since the action $(N_k, a_1)$ adds a new edge in the trajectory $\tau_{t+1}$, the size of edges in a subgraph $\mathcal{G}_1$ is equal to $|\mathcal{E}_s| + 1$. On the contrary, the action $(N_k, a_2)$ does not add any edge such that the size of edges in a subgraph $\mathcal{G}_2$ is equal to $|\mathcal{E}_s|$. Accordingly, the size of edges is different as $|\mathcal{E}(\mathcal{G}_1)| > |\mathcal{E}(\mathcal{G}_2)|$. Using the Assumption 1, their *Importance* values have the following relationship:

$$Importance(f_{unsup}(\cdot), \mathcal{G}, \mathcal{G}_1, v, k) \geq Importance(f_{unsup}(\cdot), \mathcal{G}, \mathcal{G}_2, k, v) \tag{6}$$

Therefore, $Q(N_k, a_1) \geq Q(N_k, a_2)$ is obtained. Finally, we can see that $UCB(N_k, a_1) \geq UCB(N_k, a_2)$ because of $C(N_k, a_1) = C(N_k, a_2)$. ∎

As demonstrated in Theorem 1, we have ascertained that the importance of a subgraph increases with the number of edges it contains. With the vanilla MCTS, generated subgraphs exhibit lower expressiveness, showing less preference for generating in a breadth-first manner, which requires selecting previously visited nodes, rather than a depth-first manner. Additionally, the vanilla method also carries the risk of being stuck in an isolated region with infinite iterations.

### C.2 UPPER BOUND OF *Importance* FUNCTION

We theoretically analyze the *Importance* function by deriving the upper bound.

**Theorem 2** *Assuming that $f_{unsup}(\cdot)$ is a trained unsupervised GraphSAGE model with one layer, using a mean aggregator, let $\mathcal{G}$ be an input graph, and $\mathcal{G}_s$ be a candidate explanatory subgraph for a target node $v \in \mathcal{V}$ where $\mathcal{V}$ is a list of nodes in $\mathcal{G}$. When we consider the top-k neighbors for node $v$, the Importance of the subgraph $\mathcal{G}_s$ can be bounded as follows:*

$$Importance(f_{unsup}(\cdot), \mathcal{G}, \mathcal{G}_s, v, k) \leq \frac{1}{|\mathcal{V}_{top\text{-}k}|} \sum_{u \in \mathcal{V}_{top\text{-}k}} \mathbf{C_{Lp}} ||\mathbf{M_{agg}}||_2 (||\Delta_v - \Delta_u||_2)|$$

where $\mathbf{C_{Lp}}$ is a Lipschitz constant vector for the activation function in the GraphSAGE, $\mathbf{M_{agg}}$ is a weight matrix that aggregates information from neighbors, and $\Delta_v$ is the difference between the embedding vector $f_{unsup}(v)$ (i.e., $\mathbf{emb_v}$) before and after the perturbation.

*Proof.* Since the change of top-$k$ nearest nodes in *Importance* is proportional to the change of distance of top-$k$ nearest nodes in the perturbed embedding vector, it is expressed as below:

$$Importance(f_{unsup}(\cdot), \mathcal{G}, \mathcal{G}_s, v, k) = |set(kNN(\mathbf{emb}, v, k)) - set(kNN(\mathbf{emb}', v, k))|/k$$
$$\propto \frac{1}{|\mathcal{V}_{top\text{-}k}|} \sum_{u \in \mathcal{V}_{top\text{-}k}} |dist(\mathbf{h_v}, \mathbf{h_u}) - dist(\mathbf{h'_v}, \mathbf{h'_u})| \tag{7}$$

where $\mathbf{h_v}$ means $\mathbf{emb_v}$, $\mathbf{h_u}$ means $\mathbf{emb_u}$, $\mathbf{h'_v}$ means $\mathbf{emb'_v}$, $\mathbf{h'_u}$ means $\mathbf{emb'_u}$ after perturbation for abbreviation. The top-$k$ nearest nodes $\mathcal{V}_{\text{top-}k}$ is output from the $kNN$ algorithm in the embedding space $\mathbf{emb}$. The function $dist(\mathbf{h_v}, \mathbf{h_u})$ is expressed as $\|\mathbf{h_v} - \mathbf{h_u}\|_2$ and we can define the upper bound of the importance using the Reverse triangle inequality below:

$$
\begin{aligned}
|dist(\mathbf{h_v}, \mathbf{h_u}) - dist(\mathbf{h'_v}, \mathbf{h'_u})| &= |\,\|\mathbf{h_v} - \mathbf{h_u}\|_2 - \|\mathbf{h'_v} - \mathbf{h'_u}\|_2\,| \\
&\leq \|(\mathbf{h_v} - \mathbf{h_u}) - (\mathbf{h'_v} - \mathbf{h'_u})\|_2
\end{aligned}
\tag{8}
$$

In inductive settings, we assume that $\mathbf{h_v}$ is used in the same way as GraphSAGE (Hamilton et al., 2017) with a single layer, where $\mathbf{M_{self}}$ and $\mathbf{M_{agg}}$ are trainable parameters, $\mathbf{x_v}$ is a feature of node $v$, $\mathcal{N}_v$ is a node set of node $v$'s neighbors, and $\sigma$ indicates an activation function.

$$
\mathbf{h_v} = \sigma(\mathbf{M_{self}} \mathbf{x_v} + \mathbf{M_{agg}} \frac{1}{|\mathcal{N}|} \sum_{\mathbf{u} \in \mathcal{N}_\mathbf{v}} \mathbf{x_u})
\tag{9}
$$

Next, we substitute $\mathbf{h_v}$ from Equation (8) into Equation (9). This leads us to Equations (11) and (12), obtained by canceling out the terms $\mathbf{M_{self}}\mathbf{x_v}$ and $\mathbf{M_{self}}\mathbf{x_u}$, under the premise that the node features remain unchanged. In Equation (17), we then obtain the upper bound by introducing $\mathbf{C_{Lp}}$, which is referred to as the Lipschitz constant for the activation function. By applying the Cauchy-Schwartz inequality, we derive Equation (18). In this equation, $\Delta_v$ represents the change in the embedding vector before and after perturbation, a simplification made for the sake of brevity.

$$
\|(\mathbf{h_v} - \mathbf{h_u}) - (\mathbf{h'_v} - \mathbf{h'_u})\|_2
\tag{10}
$$

$$
= \|\sigma(\mathbf{M_{self}}(\mathbf{x_v} - \mathbf{x_u}) + \mathbf{M_{agg}}(\frac{1}{|\mathcal{N}_\mathbf{v}|} \sum_{\mathbf{p} \in \mathcal{N}_\mathbf{v}} \mathbf{x_p} - \frac{1}{|\mathcal{N}_\mathbf{u}|} \sum_{\mathbf{q} \in \mathcal{N}_\mathbf{u}} \mathbf{x_q}))
\tag{11}
$$

$$
- \sigma(\mathbf{M_{self}}(\mathbf{x_v} - \mathbf{x_u}) + \mathbf{M_{agg}}(\frac{1}{|\mathcal{N'}_\mathbf{v}|} \sum_{\mathbf{p} \in \mathcal{N'}_\mathbf{v}} \mathbf{x_p} - \frac{1}{|\mathcal{N'}_\mathbf{u}|} \sum_{\mathbf{q} \in \mathcal{N'}_\mathbf{u}} \mathbf{x_q}))\|_2
\tag{12}
$$

$$
= \|\sigma(\mathbf{M_{agg}}(\frac{1}{|\mathcal{N}_v|} \sum_{p \in \mathcal{N}_v} \mathbf{x_p} - \frac{1}{|\mathcal{N}_\mathbf{u}|} \sum_{\mathbf{q} \in \mathcal{N}_\mathbf{u}} \mathbf{x_q}
\tag{13}
$$

$$
- \frac{1}{|\mathcal{N'}_v|} \sum_{p \in \mathcal{N'}_v} \mathbf{x_p} + \frac{1}{|\mathcal{N'}_\mathbf{u}|} \sum_{\mathbf{q} \in \mathcal{N'}_\mathbf{u}} \mathbf{x_q}))\|_2
\tag{14}
$$

$$
= \|\sigma(\mathbf{M_{agg}}(\frac{1}{|\mathcal{N}_v|} \sum_{p \in \mathcal{N}_v} \mathbf{x_p} - \frac{1}{|\mathcal{N'}_\mathbf{v}|} \sum_{\mathbf{p} \in \mathcal{N'}_\mathbf{v}} \mathbf{x_p}
\tag{15}
$$

$$
- (\frac{1}{|\mathcal{N}_u|} \sum_{q \in \mathcal{N}_u} \mathbf{x_q} - \frac{1}{|\mathbf{u}'|} \sum_{\mathbf{q} \in \mathcal{N}_u'} \mathbf{x_q})))\|_2
\tag{16}
$$

$$
\leq \mathbf{C_{Lp}}(\|\mathbf{M_{agg}}(\Delta_v - \Delta_u)\|_2)
\tag{17}
$$

$$
\leq \mathbf{C_{Lp}}\|\mathbf{M_{agg}}\|_2 \|\Delta_v - \Delta_u\|_2. \quad \blacksquare
\tag{18}
$$

Theorem 2 establishes that the upper bound is influenced by the weight matrices $\mathbf{M_{agg}}$, $\Delta_v$, and $\Delta_u$. Assuming that the weight matrix $\mathbf{M_{agg}}$ is fixed after training (i.e., in an inductive setting), we describe four cases of how the importance score can be changed. First, when $\Delta_v$ and $\Delta_u$ are small, the importance score becomes a low value. Second, when $\Delta_v$ is large but $\Delta_u$ is small, the target vector is relatively changed even further, causing the importance score to become large. Conversely, if $\Delta_v$ is small, but $\Delta_u$ is large, while the neighboring nodes move to a different area, the target vector remains, leading to a large importance score. Lastly, when both $\Delta_v$ and $\Delta_u$ are large, causing the vector $v$ and its top-$k$ neighboring nodes to move in the same direction, then the importance becomes small, but in the opposite direction, resulting in significant importance. Overall, the theoretical analysis suggests that the absolute value of importance is contingent on $\|\mathbf{M_{agg}}\|_2$.

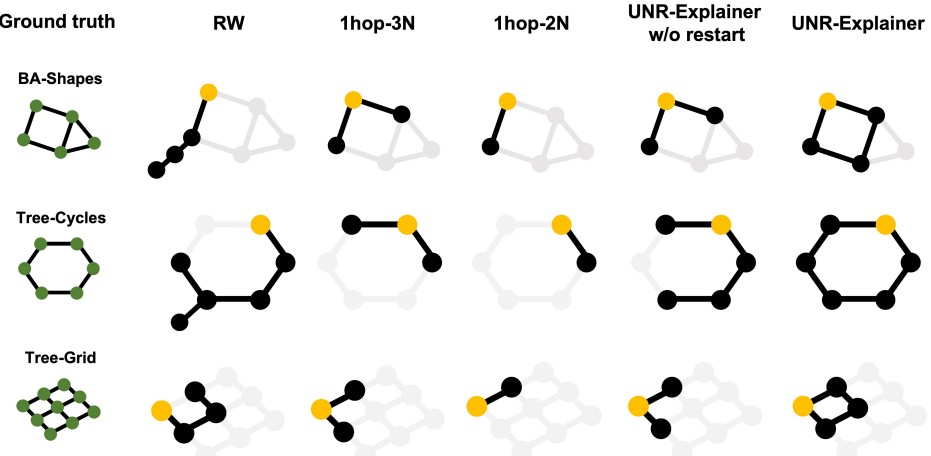

Figure 5: The examples of obtained subgraphs by different models on synthetic datasets.

# D    MORE EXPERIMENT RESULTS

## D.1    QUALITIVE ANALYSIS OF SYNTHETIC DATASETS

We conduct a case study utilizing the ground truth of the synthetic datasets. As shown in Figure 5, the yellow node represents the target node, while the black nodes and edges are parts of the subgraph obtained by each baseline model. It is important to note that the grey nodes and edges represent the $k$-hop neighboring nodes of the target node in the input graph, providing relevant contextual information. The **RW-G** method generates path-shaped subgraphs that fail to reach the ground truth, particularly noticeable on the BA-Shapes dataset. On the other hand, subgraphs generated by baselines such as **1hop-3N** and **1hop-2N** are included in the ground truth but prove insufficient when the ground truth exists beyond a 1-hop distance. In contrast, both **UNR-Explainer w/o restart** and **UNR-Explainer** more closely approximate parts of the ground truth. The difference between these two methods is not severe on small datasets, but generally, **UNR-Explainer** encourages the subgraph to have more connections centered around the target node. In conclusion, we illustrate how **UNR-Explainer** more accurately represents the ground truth, offering a superior explanation compared to other baselines.

## D.2    DETAILS OF CASE STUDY

Detailed information of nodes is described in Table 5.

## D.3    DESCRIPTION OF THE VARIANTS OF THE MCTS-BASED ALGORITHM

In Table 6, we describe the detailed information about variants of the MCTS-based algorithm presented in Table 4. The term **Action** in Table 6 refers to the design of the action in MCTS: **Adding** represents the addition of nodes, while **Removing** denotes the deletion of nodes. The term **Expansion** refers to the design of the expansion strategy in MCTS. During child node expansion, **All** represents the expansion of all neighboring nodes as child nodes, while **Sample** means that we select nodes to match the average degree of the input graph from among the neighboring nodes, thereby reducing the search space. The term **Restart** indicates whether we apply our new selection policy, as explained in Section 3. The term **UCB** refers to the variant of the UCB formula (Eq. (3)). **Prx1**, **Prx2**, and **Prx3** represent different proximity measures. **Prx1** corresponds to the degree of a node in the input graph, assuming high-degree nodes are potentially more influential. **Prx2** represents the number of common neighboring nodes of the target node in the input graph, under the assumption that nodes with more common neighbors related to the target node are potentially more influential. **Prx3** corresponds to the cosine similarity between the current state of a node and the target node in the input graph, with the assumption that nodes with similar features are potentially more influential. However, these three

Table 5: Description of nodes representing names of authors on Figure 2

| Node | Name |
|------|------|
| a | Caglar Gulcehr |
| b | Razvan Pascanu |
| c | Yann Dauphin |
| d | Yoshua Bengio |
| e | Guillaume Desjardins |
| f | Kyunghyun Cho |
| g | Surya Ganguli |
| h | Guido Montufar |
| i | Pieter Abbeel |
| j | Volodymyr Mnih |
| k | Koray Kavukcuoglu |
| l | John Schulma |
| m | Theophane Weber |
| n | Nicolas Heess |

Table 6: Description of the variants of the MCTS-based algorithm in Table 4. We abbreviate the $E$ as $\sqrt{\frac{ln(C(N_0,a))}{C(N_i,a)}}$ due to the limiation of the space.

| No. | Name | Action | Expansion | Restart | UCB |
|-----|------|--------|-----------|---------|-----|
| 1 | MCTS | Adding | All | X | $\mathrm{Avg}(W(N_i,a) + \lambda E$ |
| 2 | SubgraphX-1 | Removing | All | X | $\mathrm{Avg}(W(N_i,a)) + \lambda E$ |
| 3 | SubgraphX-2 | Removing | Sample | X | $\mathrm{Avg}(W(N_i,a)) + \lambda E$ |
| 4 | MCTS-Avg | Adding | Sample | X | $\mathrm{Avg}(W(N_i,a)) + \lambda E$ |
| 5 | MCTS-Prx1 | Adding | Sample | X | $\mathrm{Max}(W(N_i,a)) + \lambda\, Prx1\, E$ |
| 6 | MCTS-Prx2 | Adding | Sample | X | $\mathrm{Max}(W(N_i,a)) + \lambda\, Prx2\, E$ |
| 7 | MCTS-Prx3 | Adding | Sample | X | $\mathrm{Max}(W(N_i,a)) + \lambda\, Prx3\, E$ |
| 8 | UNR-Explainer w/o r | Adding | Sample | X | $\mathrm{Max}(W(N_i,a)) + \lambda E$ |
| 9 | **UNR-Explainer** | Adding | Sample | O | $\mathrm{Max}(W(N_i,a)) + \lambda\, E$ |

proximity measures did not result in significant gains, contrary to our expectations, and increased the computational cost; as a result, none of them are applied in our method.

**Comparison with SubgraphX**

The distinction between UNR-Explainer and SubgraphX is primarily discussed in terms of the reward function, action definition, and exploration strategy in MCTS. **1)** The utilization of the reward function varies, as each is designed for distinct settings. SubgraphX employs the Shapley value in the reward function, which is not suitable for unsupervised settings without labels. On the other hand, our reward metric is specifically tailored to measure the importance of subgraphs in unsupervised settings. **2)** The definition of action in UNR-Explainer differs significantly from SubgraphX. In SubgraphX, the action involves pruning nodes from the input graph until the number of remaining nodes reaches a predefined threshold. This approach is problematic when the subgraph serves as a counterfactual explanation, as more computation is required to achieve the desired size. In contrast, the action in UNR-Explainer involves adding an edge to the candidate subgraph, a more efficient strategy for exploring minimal subgraphs. The efficiency of this action is demonstrated in Table 4 which shows that SubgraphX-based methods generate larger subgraphs and require more inference time. **3)** We propose a new selection strategy, inspired by the Random walk restart, to promote more connections around the target node, suitable for our problem setting. **4)** Moreover, we employ a sampling method during the expansion stage to reduce the computational cost of exploration. Consequently, UNR-Explainer is more efficient in searching for counterfactual explanatory subgraphs from the perspective of the MCTS-based algorithm.

Table 7: The statistics of each dataset and the performance of the downstream tasks on each dataset using trained unsupervised node representation vector.

| Datasets | BA-Shapes | Tree-Cycle | Tree-Grid |
|---|---|---|---|
| # classes | 4 | 2 | 2 |
| # nodes | 700 | 871 | 1,231 |
| # edges | 2,055 | 971 | 1,565 |
| Model | GraphSAGE | GraphSAGE | GraphSAGE |
| Homogeneity | 0.314 | 0.013 | 0.015 |
| Silhouette | 0.612 | 0.998 | 0.97 |
| Task | node | node | node |
| ACC/AUC | 0.486 | 0.583 | 0.587 |
| Datasets | Cora | CiteSeer | Pubmed |
| # classes | 7 | 6 | 3 |
| # nodes | 2,708 | 3,312 | 19,717 |
| # edges | 5,429 | 4,732 | 44,338 |
| Model | GraphSAGE | GraphSAGE | DGI |
| Homogeneity | 0.471 | 0.215 | 0.006 |
| Silhouette | 0.177 | 0.160 | 0.073 |
| Task | link | node | node |
| ACC/AUC | 0.909 | 0.830 | 0.695 |

## D.4  TIME COMPLEXITY ANALYSIS

Let $t$ be the number of iterations, $n$ be the number of nodes in the search tree, $|V|$ be the number of vertices in our input graph $\mathcal{G}$, the time complexity of our UNR-Explainer w/o restart in an inductive setting mainly depends on O($t \cdot log(n) \cdot |V|$). Additionally, the $O(|V|)$ is required for the *kNN* function due to the simulation. When we employ UNR-Explainer for selection, due to the probability of the restart $p_{restart}$, the time complexity for the search is $O(n)$ in the worst case. Therefore, the total time complexity to search the important subgraph as the explanation for the target node is O($t \cdot n \cdot |V|$). In the worst case, if $n$ approaches $|V|$, the complexity could become approximately $O(|V|^2)$. To circumvent this situation, there are two strategies: establishing stopping criteria and setting an upper bound for $n$. Firstly, $t$ is set to 1,000 as a stopping criterion, or the traversal is stopped when a case meeting $Importance = 1.0$ is found. Secondly, the number of nodes in the traversal graph is limited to a constant value; in this paper, we set it to 20. Moreover, in every iteration, we sample at most 3 nodes for the expansion step, which helps avoid the exponential expansion of $n$. For these reasons, the time complexity of UNR-Explainer is approximately $O(|V|)$, because both $t$ and $n$ could be regarded as constants.

## D.5  EXPERIMENTAL SETUP

To evaluate our proposed explanation method, we first train the unsupervised node representation model using GraphSAGE (Hamilton et al., 2017) and DGI (Veličković et al., 2018). The resulting trained embedding vector is then used for downstream tasks such as link prediction and node classification. For the node classification task, we use the synthetic datasets and divide the embedding vector into a random train and test subset with 80% and 20% of the data, respectively. For the real-world datasets, we perform the link prediction task, so we split the edges of the graph into random train and test subsets with 90% and 10% of the data. Top-k link prediction is evaluated by the AUC. A logistic regression model is employed for both the node classification and the performance is evaluated by calculating the accuracy for node classification.

Additionally, to assess the quality of the initial embeddings, we report Homogeneity using labels and Silhouette Coefficient without labels, which are commonly used in clustering tasks. The homogeneity measures how instances with the same labels are contained in the same cluster, and the number of clusters is set equal to the number of classes. The Silhouette Coefficient measures how well separated each cluster is. Higher values of homogeneity and Silhouette Coefficient indicate better performance of the clustering model The above experimental setup is described in Table 7. Noteworthy, we set $k$ for top-$k$ neighbors as 5, the exploration term $\lambda$ as 1, and $p_{restart}$ as 0.2.

### D.6 IMPLEMENTATION OF DOWNSTREAM TASKS

- **Node classification**: Using the trained embedding vector in unsupervised learning, we conduct the node classification task. The logistic regression model in Scikit-learn is used. We set the max iter as 300.

- **Link prediction**: Using the trained embedding vector in unsupervised learning, we conduct the link prediction task. We split the graph to generate the train and test dataset using PyTorch-Geometric (Fey & Lenssen, 2019). To predict the link between two nodes, we assume there is a link between the target node and its top-$k$ nearest nodes.

### D.7 IMPLEMENTATION OF UNSUPERVISED NODE REPRESENTATION MODELS

We employ the GraphSAGE (Hamilton et al., 2017) and DGI (Veličković et al., 2018) model to demonstrate the performance of our proposed method in the inductive setting. We note that the GraphSAGE model and DGI are implemented using the PyTorch geometric Fey & Lenssen (2019). The detailed information is as below:

- **GraphSAGE** (Hamilton et al., 2017): We use the aggregating operator as follows: $\mathbf{x}'_i = \mathbf{M_{self}}\mathbf{x}_i + \mathbf{M_{agg}} \cdot \mathrm{mean}_{j \in \mathcal{N}_i}\mathbf{x}_j$. Here, while $\mathbf{x}$ means node features, $\mathbf{M_{self}}$ and $\mathbf{M_{agg}}$ are the trainable parameters, and $\mathcal{N}_i$ represents local neighboring nodes around node $i$. We set the hyperparameters as follows: the batch size as $256$, the number of hidden dimensions as $64$, the number of hidden layers as $2$, the dropout as $0.5$, and the optimizer as Adam. On BA-shapes, we set the number of epochs as $100$ and the learning rate as $0.01$. On other datasets, we set the number of epochs as$150$ and the learning rate as $0.01$.

- **DGI** (Veličković et al., 2018): We use the GraphSAGE as the encoder of the DGI model, setting the number of hidden dimensions as $512$, batch size as $256$, the number of hidden channels as $512$, the number of hidden layers as $2$, dropout as $0.5$, the number of epochs as $50$, optimizer as Adam, and the learning rate as $0.001$.

### D.8 BASELINES IN UNSUPERVISED LEARNING

- **Taxonomy Induction** (Liu et al., 2018): We implement the model using the code and the setting of hyperparameters from the authors in Matlab. We set the number of clusters as $50$ for all real-world datasets and $5, 3, 2$ for BA-Shapes, Tree-Cycles, and Tree-Grids respectively.

- **TAGE** (Xie et al., 2022): Since TAGE is available to provide the explanation for the embedding vector, we do not utilize the second stage which connects to the downstream task MLP to evaluate the explainer in an unsupervised manner. We set the number of epochs as $100$, the learning rate as $0.0001$, $k$ as $5$, the gradient scale as $0.2$, $\lambda_s$ as $0.05$, and $\lambda_e$ as $0.002$. The implementation is followed by (Xie et al., 2022).

### D.9 PACKAGES REQUIRED FOR IMPLEMENTATIONS

- python == 3.9.7
- pytorch == 1.13.1
- pytorch-cluster == 1.6.0
- pyg == 2.2.0
- pytorch-scatter == 2.1.0
- pytorch-sparse == 0.6.16
- cuda == 11.7.1
- numpy == 1.23.5
- tensorboardx == 2.2
- networkx == 3.0
- scikit-learn == 1.1.3
- scipy == 1.9.3
- pandas == 1.5.2

# E  LIMITATION, FUTURE WORK, AND NEGATIVE SOCIETAL IMPACTS

We focus primarily on providing counterfactual explanations for unsupervised node representation models. However, it's important to note that actionability and diversity are also key properties to consider in these explanations. A well-known example of counterfactual explanations is the case of credit application prediction discussed in (Verma et al., 2020). Counterfactual explanations offer more than mere output explanations; they also provide actionable recommendations, especially when a loan application is rejected. Additionally, providing multiple explanations enhances the diversity of possible actions, thereby allowing applicants to select the most suitable options. However, defining the desired action in unsupervised settings is challenging. Since the desired action is closely linked to the problem settings and datasets, specific assumptions, akin to the loan application example, are required. Therefore, we leave the exploration of actionability and diversity in counterfactual explanations for unsupervised models to future work. As of the current stage of our research, we have not recognized any negative societal impacts associated with this work.

