# OpenReview forum: "UNR-Explainer: Counterfactual Explanations for Unsupervised Node Representation Learning Models"
_ICLR.cc/2024/Conference — ICLR 2024 poster_

### Official Review · Reviewer_WfpS · 2023-10-22

**Soundness:** 3 good
**Presentation:** 3 good
**Contribution:** 3 good
**Rating:** 8
**Confidence:** 3

**Summary:**

The authors introduce a new method to obtain counterfactual (CF) explanations from unsupervised node learning. Their method uses a learned unsupervised node learning method to get embeddings. They then define their method via the importance function which they try to maximize while minimizing the edge alterations to the graph (minimizing the counterfactual explanation). They provide an upper bound on the Importance function.

They alter a Monte Carlo Tree Search (MCTS) method to get the CF. The MCTS uses the Importance function as a reward which looks for subgraphs that are important but sparse. They show that their MCTS does not degrade the expressiveness of subgraph explanations like the vanilla MCTS would by altering the reward function. They do this by choosing the action where the upper confidence boundary (UCB a term used in their MCTS which dictates which edge to take in traversal) is larger. They also show analysis showing that the UCB for an edge that leads to a new node is greater thus prioritizing exploring new paths. They claim these alterations to the vanilla MCTS leads to more expressive explanations which are partially supported by theoretical claims.

They also have quite a large and expansive set of experiments. They do experiments on 3 synthetic datasets (BA-Shapes, Tree-Cycles, and Tree-Grid) and 3 real datasets (Cora, CiteSeer, and PubMed), parameter sensitivity study, ablation study and a case study. They conduct their experiments on 8 methods (including theirs) on several metrics: Precision/Recall, Validity, size of the model and Importance. They show promising behaviour of their method in comparison to other methods in many settings with various evaluation metrics. They also show a case study on the NIPS dataset a social network of citations. They show by perturbing explanation graphs on a particular author they can obtain a graph that belongs to a different author that belongs to a different subfield of ML. They also conduct an ablation study showing variants of the MCTS algorithm which is fundamental to their methodology. They show their variant of MCTS can find expressive explanations (high importance score) while being efficient. Finally they also show experiments of parameter sensitivity. They show what effects the choice of the restart parameter, perturbation parameter, and the number of neighbors has on their method. They do this by varying the choice of hyperparameter and evaluating the importance score on the Cora Dataset.

**Strengths:**

The paper is very well written. The design of the paper from problem definition, to methodology, to experimental evaluation follows clearly and is well designed. The authors also motivate their work by addressing the problem in a well defined manner. The Importance measure is novel and inventive way to quantify the counterfactual explanation. The alterations to the MCTS to construct these counterfactual explanations is reasonable and well grounded by theory to supplement their decisions. The paper also employs theory on the Importance measure to show an upper bound.

The experimental list is fairly exhaustive and shows superior performance to several other methods in multiple datasets and cases. The case study is a nice touch to display their method’s ability to obtain meaningful counterfactual explanations. The ablation study shows that their variant of MCTS can find expressive subgraphs while being efficient in comparison to other tree search methods. Finally, having a study to show their methods sensitivity/robustness to choices of hyperparameters is important for anyone seeking to employ this method.

**Weaknesses:**

There could be more discussion on experiments where the UNR-Explainer underperforms compared to other methods.

Also further explanation on certain hyperparameter choices could be made more clear for the readers. Such as the choice of k in each experimental setting. A discussion on when to use a particular larger/smaller value of k would be interesting. The authors do have experiments showing the sensitivity of the number of neighbors, they also have a limited discussion on this phenomenon. However this hyperparameter is central to their method (their importance measure is heavily influenced by it) a discussion to explain what settings would require very large k vs very small k would be beneficial to solidify their work although it is not necessary.

**Questions:**

Although the authors provided a study of hyperparameter sensitivity, why did they select k=5. Clearly, as seen in the experiments the choice of k does seem to impact the importance score. More discussion of the effect of the choice of k would be beneficial for practitioners.

Also their method seems to underperform in the synthetic experiments particularly with the precision measure. Significantly smaller methods do better than UNR-Explainer in these settings which is seemingly consistent throughout the synthetic experiments. Any discussion as to why this is would be beneficial to readers and the authors.

---

> ### Author Response · Authors · 2023-11-21
> **Response to Reviewer WfpS**
>
> We sincerely appreciate your valuable feedback for reviewing this paper. For the potential concerns you bring up, we would like to address them as follows.
>
> ### **W1 \& Q2: Clarity of disscusion**
>
> Thank you for your constructive comment. In the BA-Shape dataset, there are 80 house-structured motifs, each comprising 5 nodes, which are considered the correct answers. Only a few edges in these motifs are connected to the base BA graph. In our experiments, we focused on evaluating the performance of explanation generation for nodes within these house motifs. In such cases, when employing 1hop-2N and 1hop-3N for searching explanations, the precision tends to be high. This is due to the increased probability that the explanation will include another node from the house-structured motif, thereby aligning with the ground truth. Similarly, for other synthetic datasets constructed in the same manner, 1hop-2N and 1hop-3N exhibit high precision but low recall.
>
> ### **W2 \& Q1: More discussion of choosing $k$**
>
> Thank you for your valuable question. In this paper, $k=5$ was chosen as a hyper-parameter because it performed well on average across evaluation measures such as importance, fidelity, and validity. However, we acknowledge that practitioners might select different values for $k$ depending on the desired level of interpretation. Specifically, $k$ is closely associated with the detail of the generated explanation and the number of hops, so its selection may vary based on the domain expert's judgment. For instance, in a social network analysis, a $k$  value close to the average node degree may be suitable for explanations focusing on immediate connections, such as close friends. Conversely, for understanding important local clusters, a higher $k$ might be more appropriate. This decision can be informed by domain knowledge.

---

### Official Review · Reviewer_TnSw · 2023-10-30

**Soundness:** 2 fair
**Presentation:** 2 fair
**Contribution:** 2 fair
**Rating:** 6
**Confidence:** 3

**Summary:**

The paper proposes a method called UNR-Explainer for generating counterfactual explanations in unsupervised node representation learning models. The goal of these explanations is to provide information for understanding unsupervised downstream tasks. UNR-Explainer performs Monte Carlo Tree Search to find the explanation subgraph. The subgraph importance is measured by the change of the top-k nearest neighboring nodes after perturbation. UNR-Explainer is evaluated on six datasets including both synthetic ones and real-world ones, and UNR-Explainer is shown to outperform existing explanation methods.

**Strengths:**

1. UNR-Explainer shows the good quantitative performacne, and the case study on NIPS shows UNR-Explainer can select qualititatively meaningful subgraphs.

2. The importance metric proposed in Equation 1 is novelt to me.

3. Time complexity analysis and discussion of limitation are both included in the appendix.

4. Code is provided for reproducibility.

**Weaknesses:**

1. Lacking discussions. Some baseline methods considered in the experiment section are very simple but achieve strong performance without discussion or analysis. See question 1 as well.

2. Efficiency. MCTS-based explanation can be slow than other explanation methods, e.g., gradient-based methods, especially on large graphs. This is verified by the time complexity as well.

3. Presentation can be further improved. Some figures have text that is too small to read. For example, embedding labels in figure 2.

**Questions:**

1. In Table 1 for the synthetic datasets, the naive random selection baselines 1hop-2N and 1hop-3N achieve the best results in terms of precision. Why? Any discussions?

---

> ### Author Response · Authors · 2023-11-21
> **Response to Reviewer TnSw**
>
> We sincerely appreciate your valuable feedback for reviewing this paper. For the potential concerns you bring up, we would like to address them as follows.
>
> ### **W1 \& Q1: Improving discussions**
>
> Thank you for your constructive comment. In the BA-Shape dataset, there are 80 house-structured motifs, each comprising 5 nodes, which are considered the correct answers. Only a few edges in these motifs are connected to the base BA graph. In our experiments, we focused on evaluating the performance of explanation generation for nodes within these house motifs. In such cases, when employing 1hop-2N and 1hop-3N for searching explanations, the precision tends to be high. This is due to the increased probability that the explanation will include another node from the house-structured motif, thereby aligning with the ground truth. Similarly, for other synthetic datasets constructed in the same manner, 1hop-2N and 1hop-3N exhibit high precision but low recall.
>
>
> ### **W2: Addressing efficiency**
>
> Thank you for raising this important point. UNR-Explainer has a time complexity of $O(t \cdot n \cdot |V|)$, where $t$ is the number of iterations, $n$ is the number of nodes in the search tree, and $|V|$ is the number of nodes in the input graph. In the worst case, if $n$ approaches $|V|$, the complexity could become approximately $O({|V|}^2)$. To circumvent this situation, there are two strategies: establishing stopping criteria and setting an upper bound for $n$. Firstly, $t$ is set to 1,000 as a stopping criterion, or the traversal is stopped when a case meeting $Importance = 1.0$ is found. Secondly, the number of nodes in the traversal graph is limited to a constant value; in this paper, we set it to 20. Moreover, in every iteration, we sample at most 3 nodes for the expansion step, which helps avoid the exponential expansion of $n$. For these reasons, the time complexity of UNR-Explainer is approximate $O(|V|)$, because both $t$ and $n$ could be regarded as constants. While most gradient-based methods (e.g., PG-Explainer [1]) have the time complexity of $O(|E|)$, where $|E|$ is the number of edges in the input graph, the complexity of UNR-Explainer could be more efficient than these methods, especially when the input graph is dense.
>
> In addition to addressing the time complexity issue, the MCTS-based framework finds the explanation subgraphs, which are mostly connected. In contrast, gradient-based methods such as PG-Explainer [1] incorporate a connected loss term in their objective term to achieve connectedness. Despite this, they often fail to consistently generate connected subgraphs, as the additional regularization term does not guarantee generating connected subgraphs.
>
> **Reference**
>
> [1] Luo, Dongsheng, Wei Cheng, Dongkuan Xu, Wenchao Yu, Bo Zong, Haifeng Chen, and Xiang Zhang. "Parameterized explainer for graph neural network." Advances in neural information processing systems 33 (2020): 19620-19631.
>
> ### **W3: Enhancing presentations**
> Thanks for your valuable feedback. We revised all figures to improve the presentation of our paper.

---

### Official Review · Reviewer_8JCL · 2023-10-31

**Soundness:** 3 good
**Presentation:** 2 fair
**Contribution:** 3 good
**Rating:** 6
**Confidence:** 3

**Summary:**

The paper introduces a novel method for explaining graph neural networks. The authors focus on counterfactual explanations and propose the UNR-Explainer, which aims to identify subgraphs that, when perturbed, lead to significant changes in node embeddings. The paper evaluates various explanation methods in unsupervised settings using synthetic and real-world datasets. The proposed method leverages the Monte Carlo Tree Search (MCTS) for efficient traversal in large search spaces. The paper also provides a theoretical analysis of the upper bound of Importance and discusses the algorithm for calculating Importance.

**Strengths:**

1)	The paper tackles CF reasoning in unsupervised settings, a relatively unexplored area potential implications for explainability in graph neural networks and unsupervised learning.
2)	The paper leverages the Monte Carlo Tree Search (MCTS), a technique from reinforcement learning, to efficiently traverse the search space of potential subgraphs. MCTS is known for its effectiveness in large search spaces, making it a suitable choice for this problem.
3)	The paper clearly defines the counterfactual property for unsupervised representation learning models, providing a solid foundation for their method.
4)	The paper includes a theoretical analysis of the upper bound of Importance for GraphSAGE, adding a rigorous foundation to their empirical findings.

**Weaknesses:**

1)	While the paper does evaluate on both synthetic and real-world datasets, it might benefit from testing on more diverse datasets, especially those from different domains or with different characteristics. Information on how the method scales with larger datasets or more complex graphs, and its computational efficiency, would be valuable.
2)	I believe the paper would greatly benefit from additional visual illustrations or diagrams to depict the proposed method. Visual aids can provide a clearer understanding and offer readers an intuitive grasp of the methodology. Given the complexity and novelty of the approach, diagrams or flowcharts could enhance comprehension and make the content more accessible to a broader audience.

**Questions:**

1)	How does the method scale with larger and more complex graphs? Are there any computational or memory constraints that might limit its applicability to very large datasets?
2)	How sensitive is the method to the degree of perturbation applied to the subgraph? Would minor changes in perturbation lead to significantly different results in the algorithm of importance?
3)	Given the contrastive approach employed by DGI and the inductive learning capability of GraphSAGE, how might these characteristics influence the types of counterfactual explanations generated? Furthermore, how would the proposed counterfactual explanation method adapt and perform when integrated with generative models such as GraphGAE or S2GAE?

---

> ### Author Response · Authors · 2023-11-21
> **Response to Reviewer 8JCL [1/2]**
>
> We sincerely appreciate your valuable feedback for reviewing this paper. For the potential concerns you bring up, we would like to address them as follows.
>
>
>
> ### **W1 \& Q1: Addressing scalability**
>
> Thank you for raising this important point. Scalability is an important factor to be considered particularly in real applications on very large graphs. As a matter of time complexity, UNR-Explainer has a time complexity of $O(t \cdot n \cdot |V|)$, where $t$ is the number of iterations, $n$ is the number of nodes in the search tree, and $|V|$ is the number of nodes in the input graph. In the worst case, if $n$ approaches $|V|$, the complexity could become approximately $O({|V|}^2)$. To circumvent this situation, there are two strategies: establishing stopping criteria and setting an upper bound for $n$. Firstly, $t$ is set to 1,000 as a stopping criterion, or the traversal is stopped when a case meeting $Importance = 1.0$ is found. Secondly, the number of nodes in the traversal graph is limited to a constant value; in this paper, we set it to 20. Moreover, in every iteration, we sample at most 3 nodes for the expansion step, which helps avoid the exponential expansion of $n$. For these reasons, the time complexity of UNR-Explainer is approximately $O(|V|)$, because both $t$ and $n$ could be regarded as constants. While most gradient-based methods (e.g., PGExplainer [1]) have the time complexity of $O(|E|)$, where $|E|$ is the number of edges in the input graph, the complexity of UNR-Explainer could be more efficient than these methods, especially when the input graph is dense.
>
> **Reference**
>
> [1] Luo, Dongsheng, Wei Cheng, Dongkuan Xu, Wenchao Yu, Bo Zong, Haifeng Chen, and Xiang Zhang. "Parameterized explainer for graph neural network." Advances in neural information processing systems 33 (2020): 19620-19631.
>
> ### **W2: Providing overview of UNR-Explainer**
>
> Thank you for your suggestion. We attached the figure describing an overview of UNR-Explainer in a revised version of our paper and attached [link](https://anonymous.4open.science/r/unr0929/overview.jpg).
>
>
> ### **Q2: Addressing sensitivity of perturbation effect**
> Thanks for your valuable question. As demonstrated in our additional experiments, the sensitivity of the $Importance$ measure in UNR-Explainer varies with different perturbation rates (details available at this [link](https://anonymous.4open.science/r/unr0929/perturbation_param.jpg). Specifically, when the perturbation parameter lies between 0.0 and 0.3, there is a gradual decline in $Importance$ observed on the Cora and CiteSeer datasets. However, for perturbation parameters exceeding 0.4 — implying that 40% or more of messages that ought to be weakened are retained — the effectiveness of the perturbation is reduced, leading to a lower $Importance$.

---

> ### Author Response · Authors · 2023-11-21
> **Response to Reviewer 8JCL [2/2]**
>
> ### **Q3-1: Addressing analysis of explanations by models as GraphSAGE vs DGI**
> Thanks for bringing up this question. In general, UNR-Explainer yields explanations of varying sizes when used with different node representation models such as GraphSAGE and DGI. Specifically, explanations under the GraphSAGE model average 4.5 nodes and 3.8 edges, while those under DGI average 3.9 nodes and 3.1 edges. Typically, GraphSAGE-generated explanations include more distant nodes and a greater number of connections in the input graph compared to DGI. We have provided visualizations of these explanations via this [link](https://anonymous.4open.science/r/unr0929/cora-gs-dgi.jpg). In Case 1, the explanations under GraphSAGE and DGI are common, while in Case 2, although the overall structure of the explanations is similar, the GraphSAGE explanation includes an additional node at a 2-hop distance from the target node. This trend is even more pronounced in Case 3. We hypothesize that this difference may be attributed to the methods used for defining positive and negative samples during training. GraphSAGE treats nodes near a target node on a fixed-length random walk as positive samples, as implemented in [pytorch-geometric](https://pytorch-geometric.readthedocs.io/en/latest/_modules/torch_geometric/loader/link_neighbor_loader.html#LinkNeighborLoader) with settings [10, 10]. Conversely, DGI employs a corruption function that randomly permutes node features while keeping the edge index constant to generate negative samples. This broader range of positive samples in GraphSAGE could influence UNR-Explainer to generate explanations that encompass more distant nodes and edges.
>
>
> ### **Q3-2: Adapting UNR-Explainer to explain generative models**
>
> Thank you for your insightful question. Graph Auto Encoders (GAE), such as VGAE [1] and S2GAE [2], are prominent in generation-based unsupervised and self-supervised learning on graphs. These models typically consist of two components: an encoder, which is a graph neural network producing latent representations, and a decoder, which reconstructs the input graph using these representations. The latent representation, containing implicit information, is crucial for downstream tasks like node and graph classification, where GAE models demonstrate superior performance. Hence, there is a significant demand for explaining these models in real-world applications.
>
> UNR-Explainer can be adapted to explain the encoder of GAE-based models like VGAE and S2GAE. This compatibility arises mainly because UNR-Explainer provides node-level explanations, aligning well with the output of the node-level embedding by the encoders in these models. Moreover, UNR-Explainer's ability to provide explanations without relying on class labels makes it potentially suitable for self-supervised learning models.
>
> However, adapting UNR-Explainer to GAE models involves defining explanations for generation-based GAEs and designing appropriate experimental settings, especially evaluation metrics. Additionally, the specific processes of each model must be considered. For instance, in VGAE [1], we need to control the randomness of the reparameterization trick to assess the perturbation effect accurately. In the case of S2GAE [2], which uses edge-masking and direction-aware graph masking, these perturbation methods may conflict with UNR-Explainer's perturbation process, necessitating additional adjustments. Moreover, since UNR-Explainer is designed to explain models with only an Encoder part, adapting it to generative models with a Decoder part would require further methodological considerations.
>
> To our knowledge, explainability for generative models on graphs is one of the less explored areas in research. We are grateful for your question, as it highlights an exciting direction for our future research endeavors.
>
> **Reference**
>
> [1] Kipf, Thomas N., and Max Welling. "Variational graph auto-encoders." arXiv preprint arXiv:1611.07308 (2016).
>
> [2] Tan, Qiaoyu, Ninghao Liu, Xiao Huang, Soo-Hyun Choi, Li Li, Rui Chen, and Xia Hu. "S2GAE: Self-Supervised Graph Autoencoders are Generalizable Learners with Graph Masking." In Proceedings of the Sixteenth ACM International Conference on Web Search and Data Mining, pp. 787-795. 2023.

---

> > ### Comment · Reviewer_8JCL · 2023-11-22
> > **response**
> >
> > Thank you for your comprehensive feedback. Upon reflection, I would keep my score.

---

### Official Review · Reviewer_eAe7 · 2023-10-31

**Soundness:** 3 good
**Presentation:** 3 good
**Contribution:** 2 fair
**Rating:** 6
**Confidence:** 4

**Summary:**

This work explores explanation generation for unsupervised node representation learning.  The authors propose a Monte Carlo Tree Search (MCTS)-based method to generate counterfactual (CF) explanations. Specifically, this method aims to identify the most important subgraphs that cause a significant change in the k-nearest neighbors of a node. The proposed method is incorporated into unsupervised GraphSAGE and DGI, and the performance on six datasets confirms the efficacy of the proposed method.

**Strengths:**

1. It is an interesting research topic to improve the interpretability of unsupervised learning models on graphs.
2. This can help to find the explanations of  GNN models with unseen downstream tasks.
The proposed method is tested on several datasets and shows satisfactory results.
3. The paper is well-structured and organized.

**Weaknesses:**

1. The work is somehow incremental work. SubgraphX proposed a Monte Carlo tree search algorithm to efficiently explore different subgraphs. Compared with SubgraphX, the authors seem to just add a new policy, “restart”, in the Selection step to mitigate the search bias. The design makes sense but results in limited novelty.
2. The indicators of counterfactual explanations are not rigorous. The perturbations of the input graph not only change the node embedding of interest ($emb_{v} \neq emb_{v}^{'}$) but also change other node embeddings. It does not match the Figure 1 (b) and (c) illustrated.
3. The motivation should be further improved. The authors do not state the challenges of generating counterfactual explanations in unsupervised learning compared with supervised methods, such as CF-GNNExplainer, RCExplainer, and CF2.
4. The authors do not provide real-world applications or pilot studies to support their claim that "the perturbation strategy of adding edges or nodes has a significant risk in real-world scenarios"
5. Minor error: Page 9 The first line is not left-justified; Measures in Table 1 are not arrowed.

**Questions:**

1. Why do the authors choose the MCTS-based framework rather than other gradient-based or causal-based interpretable methods? Can you show the relevant analyses?
2. Can the authors state what new challenges your approach addresses compared to existing counterfactual explanation methods on supervised learning?
3. Can the author give some applications of real-world scenarios or do some pilot studies to show the benefit of the perturbation strategy of only removing edges?

---

> ### Author Response · Authors · 2023-11-21
> **Response to Reviewer eAe7 [1/3]**
>
> We sincerely appreciate your valuable feedback for reviewing this paper. For the potential concerns you bring up, we would like to address them as follows.
>
> ### **W1: Addressing novelty**
>
> Thank you for raising the important question. We acknowledge that our method is not the first to implement Monte Carlo Tree Search (MCTS) in the graph XAI (Explainable Artificial Intelligence) domain. Prior works, such as RationaleRL [1] and SubgraphX [2], have utilized MCTS in their problem settings, each with a reward function tailored to their specific objectives. However, applying existing MCTS frameworks directly to unsupervised node learning models is challenging. The difficulty lies in defining suitable rewards and managing the increased uncertainty in exploring the graph space.
>
> To mitigate the challenges, we propose a new procedure that is different from the existing MCTS framework with respect to five key aspects. The distinctions between our model and SubgraphX are summarized in the table below.
> First, our $Importance$ function shown in Equation (1) of our main text quantifies the change of top-k neighbors in the embedding space after perturbation, bringing up the benefits of recognizing the significant influence of important subgraph $G_s$ in related downstream tasks. Thus, our contribution not only relies on developing the proper design of MCTS for counterfactual explanations but also sheds light on explaining unsupervised node representation learning. For clarification, we describe the difference between SubgraphX and UNR-Explainer regarding to the design of MCTS as below:
>
> | Design criteria        | SubgraphX          |   UNR-Explainer                |
> |------------------------|--------------------|--------------------------------|
> | Target model           | supervised models  | unsupervised models            |
> | Reward function        | Shapley value      | Importance function in Eq. (1) |
> | Action                 | to prune           | to add                         |
> | Selection              | argmax             | random walk with restart       |
> | Expansion              | without sampling   | with sampling                  |
> | Action value           | mean               | max                            |
>
> Consequently, our design of Monte Carlo Tree Search (MCTS) in UNR-Explainer reduces the computational time by 97.37\% as [(4.64 - 176.3) ÷ 176.3] ×100 and performance increased by 1.9\% as [(0.96 - 0.42) ÷ 0.942] ×100 in the Table 4 of our paper, compared to SubgraphX-1. The significant improvement in the computational cost indicates the emphasis on MCTS design to tailor its problem settings. To avoid costly and unnecessary exploration, we propose a suitable MCTS framework for counterfactual explanations in unsupervised settings. Additionally, the new selection strategy and reward functions are theoretically analyzed with respect to the bounds of value and expressiveness, respectively.
>
>  **Reference**
>
> [1] Jin, Wengong, Regina Barzilay, and Tommi Jaakkola. "Multi-objective molecule generation using interpretable substructures." In International conference on machine learning, pp. 4849-4859. PMLR, 2020.
>
> [2] Yuan, Hao, Haiyang Yu, Jie Wang, Kang Li, and Shuiwang Ji. "On explainability of graph neural networks via subgraph explorations." In International conference on machine learning, pp. 12241-12252. PMLR, 2021.
>
> ### **W2: Clarity of Figure 1**
>
> Thank you for your thoughtful comments. For brevity, we have simplified the counterfactual explanations in Figure-1 (b) and (c). As indicated, when edges near the node of interest are perturbed, the neighboring nodes' positions are affected. In contrast, nodes that are distant from the node of interest, but share structural similarities and similar feature distributions, will maintain their original embeddings unchanged. We have updated the figure to better illustrate this distinction in a revised version of our paper.

---

> ### Author Response · Authors · 2023-11-21
> **Response to Reviewer eAe7 [2/3]**
>
> ### **W3 & Q2: Addressing motivation**
>
> Thank you for pointing this out. UNR-Explainer is designed to generate counterfactual explanations in unsupervised learning contexts without relying on labels, contrasting with XAI methods in supervised learning. We highlight the limitations of existing methods when applied to unsupervised learning:
>
> - CF-GNNExplainer [1] as a pioneer of counterfactual explanation for the graph domain enhances explainability. Its loss function is formulated as
>  $L = L_{pred}(v, \bar{v}| f, g) + βL_{dist}(v,  \bar{v} | d)$, where $v$ is the original node and $\bar{v}$ is generated by the CF model. In this function, $f$ represents a predicted class label, and $g$ is the counterfactual (CF) model generating $\bar{v}$. The term $L_{pred}$ denotes a counterfactual loss ensuring $f(v) \neq f(\bar{v})$, while $L_{dist}$ quantified the distance between the input graph and the found explanation. However, as indicated in Definition 2 of our paper, the condition $f(v) \neq f(\bar{v})$ in supervised settings can be analogous to $emb_{v} \neq emb_{v}^{\prime}$ in unsupervised learning. Yet, $emb_{v} \neq emb_{v}^{\prime}$ alone provides limited information for determining the importance of each edge.  To address this and aid in generating beneficial explanations for related downstream tasks, we introduce a novel $Importance$ measure for counterfactual explanation in unsupervised node representation learning.
>
> - RCExplainer [2] tackles the robustness of counterfactual explanations by leveraging the decision boundaries of a GNN in supervised settings. However, the decision region in [2] heavily relies on class labels, making it unsuitable for unsupervised settings where class labels are absent or when no specific downstream tasks are defined.
>
> - CF2 [3] combines factual and counterfactual explanations to address the limitations of each approach. However, the process for generating explanations in CF2, as defined in Equations (10), (11), and (12) in [3], also depends on class labels for graph/node classification. This reliance makes direct application to unsupervised settings challenging.
>
> In conclusion, UNR-Explainer provides insights into counterfactual explanations in unsupervised settings without relying on class labels. It offers the advantage of describing potential impacts on related downstream tasks, such as link prediction and clustering.
>
>  **Reference**
>
> [1] Lucic, Ana, Maartje A. Ter Hoeve, Gabriele Tolomei, Maarten De Rijke, and Fabrizio Silvestri. "Cf-gnnexplainer: Counterfactual explanations for graph neural networks." In International Conference on Artificial Intelligence and Statistics*, pp. 4499-4511. PMLR, 2022.
>
> [2] Bajaj, Mohit, Lingyang Chu, Zi Yu Xue, Jian Pei, Lanjun Wang, Peter Cho-Ho Lam, and Yong Zhang. "Robust counterfactual explanations on graph neural networks."Advances in Neural Information Processing Systems 34 (2021): 5644-5655.
>
> [3] Tan, Juntao, Shijie Geng, Zuohui Fu, Yingqiang Ge, Shuyuan Xu, Yunqi Li, and Yongfeng Zhang. "Learning and evaluating graph neural network explanations based on counterfactual and factual reasoning." InProceedings of the ACM Web Conference 2022*, pp. 1018-1027. 2022.

---

> ### Author Response · Authors · 2023-11-21
> **Response to Reviewer eAe7 [3/3]**
>
> ### **W4 \& Q3: Addressing real-world scenario**
>
> Thank you for your invaluable question. We address the potential risks inherent in explainability techniques, particularly those involving modification methods like adding new nodes or edges, in real-world scenarios. In the context of social network graphs, an explanation graph generated by allowing the addition of nodes to a subgraph (used for explaining link prediction or clustering) might include nodes that are not actually observable. For example, it may suggest connections with individuals who have never met, rendering the explanation invalid for the end-user. Similarly, in citation graphs, if the explanation for link prediction includes a non-existent citation subgraph, it becomes challenging to interpret. When learning from the molecule datasets for drug discovery, permitting the addition of numerous nodes can lead to significant deviations from [chemical valency rules](https://en.wikipedia.org/wiki/Valence_(chemistry)). Such alterations could render the explanations not only inaccurate but also impractical for practical applications. To circumvent these issues, our approach emphasizes the removal of important connections to provide counterfactual explanations. This method ensures that the explanations remain grounded in the actual structure of the dataset, thereby enhancing their validity and applicability.
>
>
>
>
>
>
> ### **W5: Rectifying minor error**
>
> Thank you for your thoughtful comment. We revised this issue in the current version of our paper.
>
>
>
>
>
> ### **Q1: Rationale for MCTS**
>
> Thank you for raising the important question. Our approach offers several advantages over gradient-based or causal-based interpretable methods. Firstly, unlike gradient-based methods such as PG-Explainer [1], which attempt to ensure connectedness through a connected loss term but often fail to guarantee connected subgraphs, our MCTS-based framework reliably identifies exploratory subgraphs that are connected. This is because the additional regularization term in gradient-based methods does not consistently lead to connected subgraphs. Secondly, our objective function is not a differential equation, setting it apart from gradient-based methods. While SubgraphX [2] employs the Shapley value for the importance measurement in its MCTS framework, our $Importance$ measure quantifies the change in the k-nearest neighbor nodes after perturbation, making our MCTS-based approach more suitable for exploring significant subgraphs in unsupervised settings.
> As for causal-based methods like GEM [3] and OrphicX [4], they rely on structural causal models (SCM), which are predominantly based on the assumption of existing class labels. Developing a causal-based model for explaining embedding vectors in unsupervised settings is an intriguing area for future research. However, in the absence of external SCM models, our method solely relies on our $Importance$ function to provide explanations in unsupervised settings.
>
> **Reference**
>
> [1] Luo, Dongsheng, Wei Cheng, Dongkuan Xu, Wenchao Yu, Bo Zong, Haifeng Chen, and Xiang Zhang. "Parameterized explainer for graph neural network." Advances in neural information processing systems 33 (2020): 19620-19631.
>
> [2] Yuan, Hao, Haiyang Yu, Jie Wang, Kang Li, and Shuiwang Ji. "On explainability of graph neural networks via subgraph explorations." In International conference on machine learning, pp. 12241-12252. PMLR, 2021.
>
> [3] Lin, Wanyu, Hao Lan, and Baochun Li. "Generative causal explanations for graph neural networks." In International Conference on Machine Learning, pp. 6666-6679. PMLR, 2021.
>
> [4] Lin, Wanyu, Hao Lan, Hao Wang, and Baochun Li. "Orphicx: A causality-inspired latent variable model for interpreting graph neural networks." In Proceedings of the IEEE/CVF Conference on Computer Vision and Pattern Recognition, pp. 13729-13738. 2022.

---

> > ### Comment · Reviewer_eAe7 · 2023-11-23
> > **Response**
> >
> > Thank you for your comprehensive feedback. I would increase my score.

---

### Meta-Review · Area_Chair_TtaS · 2023-12-06

**Metareview:**

The paper describes a method called UNR-Explainer for generating counterfactual explanations in unsupervised node representation learning models. The goal is to improve the interpretability of unsupervised learning models on graphs and provide explanations for graph neural networks. UNR-Explainer utilizes MCTS to identify subgraphs that, when perturbed, lead to significant changes in node embeddings. The method is evaluated on both synthetic and real-world datasets, demonstrating its effectiveness compared to existing explanation methods.

Strengths:
* The paper addresses the relatively unexplored area of counterfactual explanations in unsupervised settings for graph neural networks.
* The paper leverages MCTS, a well-established technique from reinforcement learning, to efficiently explore the search space of potential subgraphs, making it suitable for this problem.
* The paper provides a theoretical analysis of the upper bound of Importance for GraphSAGE, adding a solid foundation to their empirical findings.
* The paper conducts extensive experiments on various datasets, including synthetic and real-world data, along with sensitivity and ablation studies, demonstrating the method's robustness and effectiveness. The case study showcases the practical utility of UNR-Explainer in real-world scenarios, illustrating its ability to provide meaningful subgraph explanations.

Weaknesses:
* Reviewers noted that the paper's contribution appears somewhat incremental compared to existing works like SubgraphX. The addition of a "restart" policy to MCTS may limit the novelty of the proposed method.
* Some reviewers suggested that the paper could benefit from additional visual aids or diagrams to clarify the methodology, given its complexity.

**Justification For Why Not Higher Score:**

The paper should address the identified weaknesses and provide more in-depth discussions and real-world applications to enhance its overall contribution and impact.

**Justification For Why Not Lower Score:**

The paper presents a promising method for generating counterfactual explanations in unsupervised node representation learning models.

---

### Decision · Program_Chairs · 2024-01-16

Accept (poster)